# An Efficient Protocol for Distributed Column Subset Selection in the Entrywise $\ell_p$ Norm

## Abstract

We give a distributed protocol with nearly-optimal communication and number of rounds for Column Subset Selection with respect to the entrywise $\ell_1$ norm ($k$-CSS$_1$), and more generally, for the $\ell_p$-norm with $1 \le p < 2$. We study matrix factorization in $\ell_1$-norm loss, rather than the more standard Frobenius norm loss, because the $\ell_1$ norm is more robust to noise, which is observed to lead to improved performance in a wide range of computer vision and robotics problems. In the distributed setting, we consider $s$ servers in the standard coordinator model of communication, where the columns of the input matrix $A \in \mathbb{R}^{d \times n}$ ($n \gg d$) are distributed across the $s$ servers. We give a protocol in this model with $\widetilde{O}(sdk)$ communication, 1 round, and polynomial running time, and which achieves a multiplicative $k^{\frac{1}{p} - \frac{1}{2}} \text{poly}(\log nd)$-approximation to the best possible column subset. A key ingredient in our proof is the reduction to the $\ell_{p,2}$-norm, which corresponds to the $p$-norm of the vector of Euclidean norms of each of the columns of $A$. This enables us to use strong coreset constructions for Euclidean norms, which previously had not been used in this context. This naturally also allows us to implement our algorithm in the popular streaming model of computation. We further propose a greedy algorithm for selecting columns, which can be used by the coordinator, and show the first provable guarantees for a greedy algorithm for the $\ell_{1,2}$ norm. Finally, we implement our protocol and give significant practical advantages on real-world data analysis tasks.

## 1 Introduction

Column Subset Selection ($k$-CSS) is a widely studied approach for rank-$k$ approximation and feature selection. In $k$-CSS, one seeks a small subset $U \in \mathbb{R}^{d \times k}$ of $k$ columns of a data matrix $A \in \mathbb{R}^{d \times n}$, typically $n \gg d$, for which there is a right factor $V$ such that $|UV - A|$ is small under some norm $|\cdot|$. $k$-CSS is a special case of low rank approximation for which the left factor is an actual subset of columns. The main advantage of $k$-CSS over general low rank approximation is that the resulting factorization is more interpretable, as columns correspond to actual features while general low rank approximation takes linear combinations of such features. In addition, $k$-CSS preserves the sparsity of the data matrix $A$.

$k$-CSS has been extensively studied in the Frobenius norm (Guruswami & Sinop, 2012; Boutsidis et al., 2014; Boutsidis & Woodruff, 2017; Boutsidis et al., 2008) and operator norms (Halko et al., 2011; Woodruff, 2014). A number of recent works (Song et al., 2017; Chierichetti et al., 2017; Dan et al., 2019; Ban et al., 2019; Mahankali & Woodruff, 2020) studied this problem in the $\ell_p$ norm ($k$-CSS$_p$) for $1 \le p < 2$. The $\ell_1$ norm is less sensitive to outliers, and better at handling missing data and non-Gaussian noise, than the Frobenius norm (Song et al., 2017). Specifically, the $\ell_1$ norm leads to improved performance in many real-world applications, such as structure-from-motion (Ke & Kanade, 2005) and image denoising (Yu et al., 2012).

Distributed low-rank approximation arises naturally when a dataset is too large to store on one machine, takes prohibitively long time for a single machine to compute a rank-$k$ approximation, or is collected simultaneously on multiple machines. Despite the flurry of recent work on $k$-CSS$_p$, this problem remains largely unexplored in the distributed setting. This should be contrasted to Frobenius norm column subset selection and low rank approximation, for which a number of results in the distributed model are known, see, e.g., Altschuler et al. (2016); Balcan et al. (2015; 2016); Boutsidis et al. (2016). We consider a widely applicable model in the distributed setting, where $s$

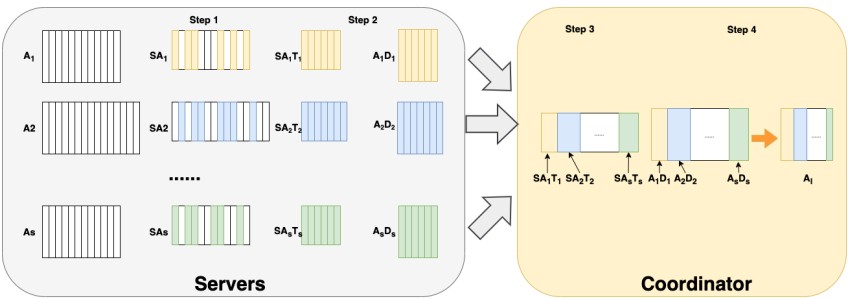

Figure 1: An overview of the proposed protocol for distributed $k$-CSS$_p$ in the column partition model. **Step 1**: Server $i$ applies a dense $p$-stable sketching matrix $S$ to reduce the row dimension of the data matrix $A_i$. $S$ is shared between all servers. **Step 2**: Server $i$ constructs a *strong coreset* for its sketched data matrix $SA_i$, which is a subsampled and reweighted set of columns of $SA_i$. Server $i$ then sends the coreset $SA_iT_i$, as well as the corresponding unsketched, unweighted columns $A_iD_i$ selected in the *strong coreset* $SA_iT_i$ to the coordinator. **Step 3**: The coordinator concatenates the $SA_iT_i$ column-wise, applies $k$-CSS$_{p,2}$ to the concatenated columns and computes the set of indices of the selected columns. **Step 4**: The coordinator recovers the set of selected columns $A_I$ from the unsketched, unweighted columns $A_iD_i$'s through previously computed indices.

servers communicate to a central coordinator via 2-way channels. This model can simulate arbitrary point-to-point communication by having the coordinator forward a message from one server to another; this increases the total communication by a factor of 2 and an additive $\log s$ bits per message to identify the destination server.

We consider the column partition model, in which each column of $A \in \mathbb{R}^{d \times n}$ is held by exactly one server. The column partition model is widely-studied and arises naturally in many real world scenarios such as federated learning (Farahat et al., 2013; Altschuler et al., 2016; Liang et al., 2014). In the column partition model, we typically have $n \gg d$, i.e., $A$ has many more columns than rows. Hence, we desire a protocol for distributed $k$-CSS$_p$ that has a communication cost that is only logarithmic in the large dimension $n$, as well as fast running time. In addition, it is important that our protocol only uses a small constant number of communication rounds (meaning back-and-forth exchanges between servers and the coordinator). Indeed, otherwise, the servers and coordinator would need to interact more, making the protocol sensitive to failures in the machines, e.g., if they go offline. Further, a 1-round protocol can naturally be adapted to an single pass streaming algorithm when we consider applications with limited memory and access to the data. In fact, our protocol can be easily extended to yield such a streaming algorithm [1].

In the following, we denote $A_{i*}$ and $A_{*j}$ as the $i$-th row and $j$-th column of $A$ respectively, for $i \in [d], j \in [n]$. We denote $A_T$ as the subset of columns of $A$ with indices in $T \subseteq [n]$. The entrywise $\ell_p$-norm of $A$ is $|A|_p = (\sum_{i=1}^{d} \sum_{j=1}^{n} |A_{ij}|^p)^{\frac{1}{p}}$. The $\ell_{p,2}$ norm is defined as $|A|_{p,2} = (\sum_{j=1}^{d} |A_{*j}|_2^p)^{\frac{1}{p}}$. We consider $1 \le p < 2$. We denote the best rank-$k$ approximation error for $A$ in $\ell_p$ norm by **OPT** $:= \min_{\text{rank-k } A_k} |A - A_k|_p$. Given an integer $k > 0$, we say $U \in \mathbb{R}^{d \times k}$, $V \in \mathbb{R}^{k \times n}$ are the left and right factors of a rank-$k$ factorization for $A$ in the $\ell_p$ norm with approximation factor $\alpha$ if $|UV - A|_p \le \alpha \cdot$ **OPT**.

Since general rank-$k$ approximation in $\ell_1$ norm is NP hard (Gillis & Vavasis, 2015), we follow previous work and consider bi-criteria $k$-CSS algorithms which obtain polynomial running time. Instead of outputting exactly $k$ columns, such algorithms return a subset of $\widetilde{O}(k)$ columns of $A$, suppressing logarithmic factors in $k$ or $n$. It is known that the best approximation factor to **OPT** that can be obtained through the span of a column subset of size $\widetilde{O}(k)$ is $\Omega(k^{1/2-\gamma})$ for $p = 1$ (Song et al., 2017) and $\Omega(k^{1/p-1/2-\gamma})$ for $p \in (1, 2)$ (Mahankali & Woodruff, 2020), where $\gamma$ is an arbitrarily small constant.

---

[1]We provide a detailed analysis of our streaming algorithm in Appendix E.

## 1.1 Previous Approaches to $k$-CSS$_p$ in the Distributed Setting

If one only wants to obtain a good left factor $U$, and not necessarily a column subset of $A$, in the column partition model, one could simply sketch the columns of $A_i$ by applying an oblivious sketching matrix $S$ on each server. Each server sends $A_i \cdot S$ to the coordinator. The coordinator obtains $U = AS$ as a column-wise concatenation of $A_i S$'s. Song et al. (2017) showed that $AS$ achieves an $\widetilde{O}(\sqrt{k})$ approximation to **OPT**, and this protocol only requires $\widetilde{O}(sdk)$ communication, $O(1)$ rounds and polynomial running time. However, while $AS$ is a good left factor, it does not correspond to an actual subset of columns of $A$.

Obtaining a subset of columns that approximates $A$ well with respect to the $p$-norm in a distributed setting is non-trivial. One approach due to Song et al. (2017) is to take the matrix $AS$ described above, sample rows according to the Lewis weights (Cohen & Peng, 2015) of $AS$ to get a right factor $V$, which is in the row span of $A$, and then use the Lewis weights of $V$ to in turn sample columns of $A$. Unfortunately, this protocol only achieves a loose $\widetilde{O}(k^{3/2})$ approximation to **OPT** (Song et al., 2017). Moreover, it is not known how to do Lewis weight sampling in a distributed setting. Alternatively, one could adapt existing single-machine $k$-CSS$_p$ algorithms to the distributed setting under the column partition model. Existing works on polynomial time $k$-CSS$_p$ (Chierichetti et al., 2017; Song et al., 2019b; Dan et al., 2019; Mahankali & Woodruff, 2020) give bi-criteria algorithms, and are based on a recursive framework with multiple rounds, which is as follows: in each round, $\widetilde{O}(k)$ columns are selected uniformly at random, and with high probability, the selected columns can provide a good approximation to a constant fraction of all columns of $A$. Among the remaining columns that are not well approximated, $\widetilde{O}(k)$ columns are recursively selected until all columns of $A$ are well approximated, resulting in a total of $O(\log n)$ rounds.

A naïve extension of this bi-criteria $k$-CSS$_p$ framework to a distributed protocol requires $O(\log n)$ rounds, as in each round, the servers and the coordinator need to communicate with each other in order to find the columns that are covered well and select from the remaining unselected columns. To reduce this to a single round, one might consider running the $O(\log n)$ round selection procedure on the coordinator only. In order to do this, the coordinator needs to first collect all columns of $A$ from the servers, but directly communicating all columns is prohibitive.

Alternatively, one could first apply $k$-CSS$_p$ on $A_i$ to obtain factors $U_i$ and $V_i$ on each server, and then send the coordinator all of the $U_i$ and $V_i$. The coordinator then column-wise stacks the $U_i V_i$ to obtain $U \cdot V$ and selects $\widetilde{O}(k)$ columns from $U \cdot V$. Even though this protocol applies to all $p \geq 1$, it achieves a loose $O(k^2)$ approximation to **OPT** and requires a prohibitive $O(n + d)$ communication cost[2]. One could instead try to just communicate the matrices $U_i$ to the coordinator, which results in much less communication, but this no longer gives a good approximation. Indeed, while each $U_i$ serves as a good approximation locally, there may be columns that are locally not important, but become globally important when all of the matrices $A_i$ are put together. What is really needed here is a small *coreset* $C_i$ for each $A_i$ so that if one concatenates all of the $C_i$ to obtain $C$, *any* good column subset of the coreset $C$ corresponds to a good column subset for $A$. Unfortunately, coresets for the entrywise $\ell_p$-norm are not known to exist.

## 1.2 Our Contributions

**Our Distributed Protocol** We overcome these problems and propose the first efficient protocol for distributed $k$-CSS$_p$ ($1 \leq p < 2$) in the column partition model that selects $\widetilde{O}(k)$ columns of $A$ achieving an $\widetilde{O}(k^{1/p-1/2})$-approximation to **the best possible subset of columns** and requires only $\widetilde{O}(sdk)$ communication cost, 1 round and polynomial time. Figure 1 gives an overview of the protocol. We note that our subset of columns does not necessarily achieve an $\widetilde{O}(k^{1/p-1/2})$-approximation to **OPT** itself, although it does achieve such an approximation to the best possible subset of columns. Using the fact that there always exists a subset of columns providing an $\widetilde{O}(k^{1/p-1/2})$-approximation to **OPT** (Song et al., 2017), we conclude that our subset of columns achieves an $\widetilde{O}(k^{2/p-1})$-approximation to **OPT**. Recently, and independently of our work, Mahankali & Woodruff (2020) show how to obtain a subset of columns achieving an $\widetilde{O}(k^{1/p-1/2})$-approximation to **OPT** itself; however, such a subset is found by uniformly sampling columns in $O(\log n)$ *adaptive* rounds using the recursive sampling

---

[2]We give this protocol and the analysis in Appendix A

framework above, and is inherently hard to implement in a distributed setting with fewer rounds. In contrast, our protocol achieves 1 round of communication, which is optimal.

We make use of a strong *coreset*, i.e., a sampled and reweighted subset of columns of each $A_i$ that approximates the cost of all potential left factors of $A_i$, by first embedding all subspaces spanned by *any* subset of $\widetilde{O}(k)$ columns of $A$ from $\ell_p$-space to Euclidean space, to bypass the lack of strong *coresets* for the $\ell_p$ norm. We denote this new norm as $\ell_{p,2}$ norm, which is the sum of the $p$-th powers of the $\ell_2$ norms of the columns. To reduce the error incurred by switching to the $\ell_{p,2}$-norm, we reduce the row dimension of $A$ by left-multiplying by an oblivious sketching matrix $S$ shared across servers, resulting in an overall approximation factor of only $\widetilde{O}(k^{1/p-1/2})$. Afterwards, each server sends its strong coreset to the coordinator. The coordinator, upon receiving the coresets from each server, runs an $O(1)$-approximate bi-criteria $k$-CSS$_{p,2}$ algorithm to select the final column subset, giving an overall $\widetilde{O}(k^{1/p-1/2})$ approximation to the best column subset.

We introduce several new technical ideas in the analysis of our protocol. Our work is the first to apply a combination of oblivious sketching in the $p$-norm via $p$ stable random variables and strong *coresets* in the $\ell_{p,2}$ norm (Sohler & Woodruff, 2018; Huang & Vishnoi, 2020) to distributed $k$-CSS. Furthermore, to show that our oblivious sketching step only increases the final approximation error by a logarithmic factor, we combine a net argument with a union bound over all possible subspaces spanned by column subsets of $A$ of size $\widetilde{O}(k)$. Previous arguments involving sketching, such as those by Song et al. (2017); Ban et al. (2019); Mahankali & Woodruff (2020), only consider a single subspace at a time.

**Theoretical Guarantees and Empirical Benefits for Greedy $k$-CSS$_{1,2}$** We also propose a greedy algorithm to select columns in the $k$-CSS$_{1,2}$ step of our protocol, and show the first additive error guarantee compared to **the best possible subset $A_S$ of columns**, i.e., our cost is at most $(1 - \epsilon) \min_V |A_S V - A|_{1,2} + \epsilon |A|_{1,2}$. Similar error guarantees were known for the Frobenius norm (Altschuler et al., 2016), though nothing was known for the $\ell_{1,2}$ norm. We also implement our protocol and experiment with distributed $k$-CSS$_1$ on various real-world datasets. We compare the $O(1)$-approximate bi-criteria $k$-CSS$_{1,2}$ and the greedy $k$-CSS$_{1,2}$ as different possible subroutines in our protocol, and show that greedy $k$-CSS$_{1,2}$ yields an improvement in practice.

## 2 PROBLEM SETTING

### 2.1 THE COLUMN PARTITION MODEL

We consider a model where there are $s$ servers, the $i^{th}$ of which holds $A_i \in \mathbb{R}^{d \times n_i}$, and a coordinator which initially does not hold any data. Each server talks only to the coordinator, via a 2-way communication channel. The communication cost is the total number of words transferred between the servers and the coordinator over the course of the protocol. Each word is $O(\log(snd))$ bits. The overall data matrix $A \in \mathbb{R}^{d \times n}$ is $A = [A_1, A_2, \ldots, A_s]$ (the column-wise concatenation of the $A_i$'s). Here, $n$ is defined to be $\sum_{i=1}^{s} n_i$. Typically, in the column partition model, $n \gg d$.

### 2.2 $p$-STABLE DISTRIBUTION AND $p$-STABLE RANDOM VARIABLES

Let $Z, X_1, \ldots, X_d$ be random variables drawn i.i.d. from some distribution $\mathcal{D}$. $\mathcal{D}$ is called $p$-stable if for an arbitrary vector $v \in \mathbb{R}^d$, $\langle v, X \rangle = |v|_p Z$, where $X = [X_1, \ldots, X_d]^T$. Random variables drawn from such a distribution are called $p$-stable random variables, which exist for $p \in (0, 2]$. Though there is no closed form expression for the $p$-stable distribution in general except for a few values of $p$, we can efficiently generate a single $p$-stable random variable in $O(1)$ time using the following method due to Chambers et al. (1976): if $\theta \in [-\frac{\pi}{2}, \frac{\pi}{2}]$ and $r \in [0, 1]$ are sampled uniformly at random, then, $\frac{\sin(p\theta)}{\cos^{1/p}\theta} \left( \frac{\cos(\theta(1-p))}{\ln(\frac{1}{r})} \right)^{\frac{1-p}{p}}$ follows a $p$-stable distribution.

## 3 PRELIMINARIES FOR OUR PROTOCOL

We first note a standard relationship between the $\ell_p$ norm and the $\ell_{p,2}$ norm.

**Lemma 1.** *For a matrix $A \in \mathbb{R}^{d \times n}$ and $p \in [1, 2)$, $|A|_{p,2} \leq |A|_p \leq d^{\frac{1}{p} - \frac{1}{2}} |A|_{p,2}$.*

### 3.1 $\ell_p$-NORM OBLIVIOUS SKETCHING

We left-multiply $A$ by an oblivious sketching matrix $S$ with $p$-stable random variables so that we lose only an $\widetilde{O}(k^{\frac{1}{p}-\frac{1}{2}})$ approximation factor when we switch to the $\ell_{p,2}$ norm.

The purpose of the next two lemmas is to show that we can perform oblivious sketching while preserving the costs of *all* possible column subsets up to logarithmic factors. We first show a lower bound on the approximation error for a sketched subset of columns, $|SA_T V - SA|_p$, which holds simultaneously for any arbitrary subset $A_T$ of chosen columns, and for any arbitrary right factor $V$.

**Lemma 2** (Sketched Error Lower Bound). *Let $A \in \mathbb{R}^{d \times n}$ and $k \in \mathbb{N}$. Let $t = k \cdot poly(\log(nd))$, and let $S \in \mathbb{R}^{t \times d}$ be a matrix whose entries are i.i.d. standard p-stable random variables, rescaled by $\Theta(1/t^{\frac{1}{p}})$. Then, with probability $1 - o(1)$, for all $T \subset [n]$ with $|T| = k \cdot poly(\log k)$ and for all $V \in \mathbb{R}^{|T| \times n}$, $|A_T V - A|_p \leq |SA_T V - SA|_p$.*

Next, we show an upper bound on the approximation error of $k$-$CSS_p$ on a sketched subset of columns, $|SA_T V - SA|_p$, which holds for a *fixed* subset of columns $A_T$ and a *fixed* right factor $V$.

**Lemma 3** (Sketched Error Upper Bound). *Let $A \in \mathbb{R}^{d \times n}$ and $k \in \mathbb{N}$. Let $t = k \cdot poly(\log(nd))$, and let $S \in \mathbb{R}^{t \times d}$ be a matrix whose entries are i.i.d. standard p-stable random variables, rescaled by $\Theta(1/t^{\frac{1}{p}})$. Then, for a fixed subset $T \subset [n]$ of columns with $|T| = k \cdot poly(\log k)$ and a fixed $V \in \mathbb{R}^{|T| \times n}$, with probability $1 - o(1)$, we have $\min_V |SA_T V - SA|_p \leq \min_V O(\log^{1/p}(nd))|A_T V - A|_p$.*

### 3.2 STRONG CORESETS IN THE $\ell_{p,2}$ NORM

To enable sub-linear communication cost in the number $n$ of columns, the $i$-th server sends the coordinator a strong coreset of columns of $SA_i$, which is a reweighted subset of the columns of $SA_i$. Such strong coresets preserve the error incurred by any rank-$k$ projection, up to a constant factor, in the $\ell_{p,2}$ norm. The coreset of a matrix $A \in \mathbb{R}^{d \times n}$ is usually denoted as $AT$, where $T = D \cdot W \in \mathbb{R}^{n \times t}$ is a sampling and reweighting matrix and $t$ is the number of columns to be included in the coreset. The sampling matrix $D$ is a matrix with $t$ columns where each column has only one 1 in the index of the column of $A$ to be included in the coreset and 0 everywhere else. The reweighting matrix $W$ is a diagonal $t \times t$ matrix with weights associated with each sample in the coreset.

**Lemma 4** (Strong Coreset in $\ell_{p,2}$ norm). *Let $A \in \mathbb{R}^{d \times n}$, $k \in \mathbb{N}$, $p \in [1, 2)$, and $\epsilon, \delta \in (0, 1)$. Then, in $n \cdot poly(k \log n/\epsilon)$ time, one can find a sampling and reweighting matrix $T$ with $O(d \log d/\epsilon^2) \cdot \log(1/\delta)$ columns such that, with probability $1 - \delta$, for all rank-$k$ matrices $U$,*

$$\min_{rank\text{-}k\ V} |UV - AT|_{p,2} = (1 \pm \epsilon) \min_{rank\text{-}k\ V} |UV - A|_{p,2}$$

*$AT$ is called a **strong coreset** of $A$.*

### 3.3 POLYNOMIAL TIME, $O(1)$-APPROXIMATE BI-CRITERIA $k$-$CSS_{p,2}$

After server $i$ sends a strong coreset to the coordinator, the coordinator does $k$-$CSS$ on a column-wise concatenation of these coresets, in the $\ell_{p,2}$ norm rather than the $\ell_p$ norm. We give a polynomial time, $O(1)$-approximate bi-criteria $k$-$CSS_{p,2}$ algorithm for $p \in [1, 2)$.

**Theorem 5** (Bicriteria $O(1)$-Approximation Algorithm for $k$-$CSS_{p,2}$). *Let $A \in \mathbb{R}^{d \times n}$ and $k \in \mathbb{N}$. There exists an algorithm that runs in $(nnz(A) + d^2) \cdot kpoly(\log k)$ time and outputs a rescaled subset of columns $U \in \mathbb{R}^{d \times \widetilde{O}(k)}$ of $A$ and a right factor $V \in \mathbb{R}^{\widetilde{O}(k) \times n}$ for which $V = \min_V |UV - A|_{p,2}$, such that with probability $1 - o(1)$,*

$$|UV - A|_{p,2} \leq O(1) \cdot \min_{rank\text{-}k\ A_k} |A_k - A|_{p,2}$$

Our polynomial time bi-criteria $k$-$CSS_{p,2}$ algorithm is based on that of Clarkson & Woodruff (2015). The main difference is that the algorithm of Clarkson & Woodruff (2015) outputs a subset with $O(k^2)$ columns due to the usage of $\ell_p$ leverage scores — we reduce the number of selected columns to $\widetilde{O}(k)$ by using $\ell_p$ Lewis weights (Cohen & Peng, 2015). Details are given in Appendix C.

---

**Algorithm 1** An efficient protocol for bi-criteria $k$-$\mathrm{CSS}_p$ in the column partition model

---

**Initial State:** Server $i$ holds matrix $A_i \in \mathbb{R}^{d \times n_i}, \forall i \in [s]$.
**Coordinator:**
Generate a dense $p$-stable sketching matrix $S \in \mathbb{R}^{k \operatorname{poly}(\log(nd)) \times d}$.
Send $S$ to all servers.
**Server $i$:**
Compute $SA_i$.
Let the number of samples in the coreset be $t = O(k\operatorname{poly}(\log(nd)))$. Construct a coreset of $SA_i$ under the $\ell_{p,2}$ norm by applying a sampling matrix $D_i$ of size $n_i \times t$ and a diagonal reweighting matrix $W_i$ of size $t \times t$.
Let $T_i = D_i W_i$. Send $SA_i T_i$ along with $A_i D_i$ to the coordinator.
**Coordinator:**
Column-wise stack $SA_i T_i$ to obtain $SAT = [SA_1 T_1, SA_2 T_2, \ldots, SA_s T_s]$.
Apply $k$-$\mathrm{CSS}_{p,2}$ on $SAT$ to obtain the indices $I$ of the subset of selected columns with size $O(k \cdot \operatorname{poly}(\log k))$.
Since $D_i$'s are sampling matrices, the coordinator can recover the original columns of $A$ by mapping indices $I$ to $A_i D_i$'s.
Denote the final selected subset of columns by $A_I$. Send $A_I$ to all servers.
**Server $i$:**
Solve $\min_{V_i} |A_I V_i - A_i|_p$ to obtain the right factor $V_i$. $A_I$ and $V$ will be factors of a rank-$k \cdot \operatorname{poly}(\log k)$ factorization of $A$, where $V$ is the (implicit) column-wise concatenation of the $V_i$.

---

## 4    AN EFFICIENT PROTOCOL FOR DISTRIBUTED $k$-$\mathrm{CSS}_p$

**Theorem 6** (A Protocol for Distributed $k$-$\mathrm{CSS}_p$). *In the column partition model, let $A \in \mathbb{R}^{d \times n}$ be the data matrix whose columns are partitioned across $s$ servers and suppose server $i$ holds a subset of columns $A_i \in \mathbb{R}^{d \times n_i}$, where $n = \sum_{i \in [s]} n_i$. Then, given $p \in [1, 2)$ and a desired rank $k \in \mathbb{N}$, Algorithm 1 outputs a subset of columns $A_I \in \mathbb{R}^{d \times k\operatorname{poly}(\log(k))}$ in $\widetilde{O}(\operatorname{nnz}(A)k + kd + k^3)$ time, such that with probability $1 - o(1)$,*

$$\min_V |A_I V - A|_p \leq \widetilde{O}(k^{1/p - 1/2}) \min_{L \subset [n], |L| = k} |A_L V - A|_p$$

*Algorithm 1 uses $1$ round of communication and $\widetilde{O}(sdk)$ words of communication.*

*Proof.* **Approximation Factor.**    In the following proof, let $L \subset [n], |L| = k$ denote the best possible subset of $k$ columns of $A$ that gives the minimum $k$-$\mathrm{CSS}_p$ cost, i.e., the cost $\min_V |A_L V - A|_p$ achieves minimum. First, note that

$$\min_V |A_I V - A|_p$$
$$\leq |A_I V' - A|_p \qquad\qquad\qquad\qquad V' := \arg\min_V |SA_I V - SA|_{p,2}$$
$$\leq |SA_I V' - SA|_p \qquad\qquad\qquad\qquad \text{By \textbf{Lemma} 2}$$
$$= \widetilde{O}(k^{\frac{1}{p} - \frac{1}{2}})|SA_I V' - SA|_{p,2} \qquad \text{By \textbf{Lemma} 1, and } S \text{ has } k \cdot \operatorname{poly}(\log(nd)) \text{ rows}$$

$SA_I$ is the selected columns output from the bi-criteria $O(1)$-approximation $k$-$\mathrm{CSS}_{p,2}$ algorithm. Let $(SAT)^*$ denote the best rank $k$ approximation to $SAT$. By **Theorem 5**,

$$\widetilde{O}(k^{\frac{1}{p} - \frac{1}{2}})|SA_I V' - SA|_{p,2} \leq \widetilde{O}(k^{\frac{1}{p} - \frac{1}{2}}) \cdot O(1)|(SAT)^* - SAT|_{p,2}$$
$$\leq \widetilde{O}(k^{\frac{1}{p} - \frac{1}{2}}) \min_V |SA_L V - SAT|_{p,2}$$

Note that $SAT = [SA_1 T_1, \ldots, SA_s T_s]$ is a column-wise concatenation of all coresets of $SA_i$, $\forall i \in [s]$. By **Lemma 4**,

$$(\min_V |SA_L V - SAT|_{p,2}^p)^{1/p} = (\sum_{i=1}^s \min_{V_i} |SA_L V_i - SA_i T_i|_{p,2}^p)^{1/p}$$

$$= (\sum_{i=1}^{s} (1 \pm \epsilon)^p \min_{V_i} |SA_L V_i - SA_i|_{p,2}^p)^{1/p} = (1 \pm \epsilon)(\sum_{i=1}^{s} \min_{V_i} |SA_L V_i - SA_i|_{p,2}^p)^{1/p}$$

$$= (1 \pm \epsilon) \min_V |SA_L V - SA|_{p,2}$$

Hence,

$$\widetilde{O}(k^{\frac{1}{p}-\frac{1}{2}}) \min_V |SA_L V - SAT|_{p,2} \leq \widetilde{O}(k^{\frac{1}{p}-\frac{1}{2}}) \min_V |SA_L V - SA|_{p,2}$$

$$\leq \widetilde{O}(k^{\frac{1}{p}-\frac{1}{2}}) \min_V |SA_L V - SA|_p \qquad \text{By \textbf{Lemma} 1}$$

$$\leq \widetilde{O}(k^{\frac{1}{p}-\frac{1}{2}}) \cdot \log^{1/p}(nd) \min_V |A_L V - A|_p \quad \text{By \textbf{Lemma} 3}$$

Therefore, we conclude: $\min_V |A_I V - A|_p \leq \widetilde{O}(k^{\frac{1}{p}-\frac{1}{2}}) \min_V |A_L V - A|_p$.

**Communication Cost.** Sharing the dense $p$-stable sketching matrix $S$ with all servers costs $O(sdk \cdot \text{poly}(\log(nd)))$ communication (this can be removed with a shared random seed). Sending all coresets $SA_i T_i$ ($\forall i \in [s]$) and the corresponding columns $A_i D_i$ to the coordinator costs $\widetilde{O}(sdk)$ communication, since each coreset contains only $\widetilde{O}(k)$ columns. Finally, the coordinator needs $\widetilde{O}(sdk)$ words of communication to send the $\widetilde{O}(k)$ selected columns to each server. Therefore, the overall communication cost is $\widetilde{O}(sdk)$, suppressing a logarithmic factor in $n, d$.

**Running time.** Since generating a single $p$-stable random variable takes $O(1)$ time, generating the dense $p$-stable sketching matrix $S$ takes $O(dk \cdot \text{poly}(\log(nd)))$ time. Computing all $SA_i$'s takes $O(\text{nnz}(A)k \cdot \text{poly}(\log(nd)))$ time. By Lemma 4, computing a single coreset $SA_i T_i$ on server $i$ takes time $n_i \text{poly}(k \log n_i)$ and thus computing all coresets across the servers takes at most time $n\text{poly}(k \log n)$. By Theorem 5, the $k$-CSS$_{p,2}$ algorithm takes time $(\text{nnz}(SAT) + k^2\text{poly}(\log nd)) \cdot k\text{poly}(\log k) \leq k^3\text{poly}(\log(knd))$ to find the set of selected columns. Since the number of selected columns is $O(k\text{poly}(\log k))$, it then takes the protocol $O(k\text{poly}(\log k))$ time to map the indices of the output columns from $k$-CSS$_{p,2}$ to recover the original columns $A_I$. Therefore, the overall running time for the protocol to find the subset of columns $A_I$ is $\widetilde{O}((\text{nnz}(A)k + kd + k^3))$, suppressing a low degree polynomial dependency on $\log(knd)$. After the servers receive $A_I$, it is possible to solve $\min_{V_i} |A_I V_i - A_i|_p$ in $\widetilde{O}(\text{nnz}(A_I)) + \text{poly}(d \log n)$ time, $\forall i \in [s]$ due to Wang & Woodruff (2019); Yang et al. (2018). $\square$

## 5 GREEDY $k$-CSS$_{1,2}$

We propose a greedy algorithm, shown in Algorithm 2, for $k$-CSS$_{1,2}$, which can be used in the place of the algorithm described in Theorem 5. The basic version of this algorithm, discussed in Appendix D, performs $k$-CSS$_{1,2}$ by simply selecting the additional column, among those of $A$, that reduces the approximation error the most at each iteration. Our analysis of that algorithm is inspired by the analysis of Greedy $k$-CSS$_2$ for the Frobenius norm in Altschuler et al. (2016). Here we provide the first additive error guarantee, compared to the best possible subset of columns, for the greedy $k$-CSS$_{1,2}$ algorithm. For a faster running time, we make use of the Lazier-than-lazy heuristic described in Section 5.2 of Altschuler et al. (2016), where in each iteration, rather than considering all columns of $A$ as candidate additional columns of $A_T$, we only sample a subset of the columns of $A$ of size $O(\frac{n \log(1/\delta)}{k})$, and pick the column among those that improves the objective the most.

**Theorem 7.** *Let $A \in \mathbb{R}^{d \times n}$ be the data matrix and $k \in \mathbb{N}$ be the desired rank. Let $A_S$ be the best possible subset of $k$ columns, i.e., $A_S = \arg\min_{A_S} \min_V |A_S V - A|_{1,2}$. Let $\sigma$ be the minimum non-zero singular value of the matrix $B$ of normalized columns of $A_S$, (the $j$-th column of $B$ is $B_{*j} = (A_S)_{*j}/|(A_S)_{*j}|_2$). Then, if $T \subset [n]$ is the subset of columns selected by Greedy $k$-CSS$_{1,2}$, the following holds with $|T| = \Omega(\frac{k}{\sigma^2 \epsilon^2})$:*

$$\min_V |A_T V - A|_{1,2} \leq (1 - \epsilon) \min_{S \subset [n], |S|=k, V \in \mathbb{R}^{k \times n}} |A_S V - A|_{1,2} + \epsilon |A|_{1,2}$$

*Similarly, if $T \subset [n]$ is the subset of columns selected by Lazier-than-lazy Greedy $k$-CSS$_{1,2}$, the following holds with $|T| = \Omega(\frac{k}{\sigma^2 \epsilon^2})$ and $\delta = \epsilon$:*

$$E[\min_V |A_T V - A|_{1,2}] \leq (1 - \epsilon) \min_{S \subset [n], |S|=k, V \in \mathbb{R}^{k \times n}} |A_S V - A|_{1,2} + \epsilon |A|_{1,2}$$

---

**Algorithm 2** Lazier-than-lazy Greedy $k$-CSS$_{1,2}$. This version of the greedy algorithm is based on Section 5.2 of Altschuler et al. (2016).

---

**Input:** The data matrix $A \in \mathbb{R}^{d \times n}$, the number of iterations $r \le n$, a parameter $\delta \in (0, 1)$.
**Output:** A subset of columns $A_T$ from $A$, where $|T| = r$.
$A_T \leftarrow \emptyset$
**for** $i = 1$ **to** $r$ **do**
$\quad \overline{T} \leftarrow$ A subset of $\frac{n \log(1/\delta)}{k}$ columns of $A$, each selected uniformly at random (excluding the columns whose indices are in $T$)
$\quad$ Column $j^* \leftarrow \arg \min_{j \in \overline{T}} (\min_V |A_{T \cup j} V - A|_{1,2})$
$\quad A_T \leftarrow A_{T \cup j^*}$
**end for**

---

| Dataset | Size | # servers $s$ | Column Distribution | Rank $k$ |
|---------|------|---------------|---------------------|----------|
| synthetic | $(2000 + k) \times (2000 + k)$ | 2 | 1001, 1002 | $\{10, 20, 30\}$ |
| TechTC | $139 \times 18446$ | 20 | 922 columns on 19 servers, 928 columns on 1 server | $\{10, 30, 50, 70, 100\}$ |

Table 1: A summary of datasets used in the experiments.

*If we let $|T| = \Omega(\frac{k}{\sigma^2 \epsilon^2})$, then the overall running time of Algorithm 2 is $O(\frac{n \log(1/\epsilon)}{\sigma^2 \epsilon^2} \cdot F)$, where $F$ is the running time needed to evaluate $\min_V |A_{T \cup j} V - A|_{1,2}$ for a fixed $j \in \overline{T}$. We can get $F = O(\frac{dk^2}{\sigma^4 \epsilon^4} + \frac{ndk}{\sigma^2 \epsilon^2})$ by taking $A_{T \cup j}^{\dagger}$.*

Since the error upper bound for greedy $k$-CSS$_{p,2}$ depends on $|A|_{1,2}$, it is not directly comparable to the error upper bound for the proposed $k$-CSS$_{p,2}$ from Subsection 3.3, which achieves a multiplicative $O(1)$-approximation to the best rank-$k$ approximation. We empirically compare the two versions of $k$-CSS$_{p,2}$ for $p = 1$ in Section 6.

## 6 EXPERIMENTS

We implement our protocol for distributed $k$-CSS$_p$ in Algorithm 1, setting $p = 1$, which enables us to compare two subroutines on the coordinator: Regular $k$-CSS$_{1,2}$ from Section 3.3 and Greedy $k$-CSS$_{1,2}$ from Section 5. We compare our $k$-CSS$_1$ protocol against a commonly applied baseline for $\ell_p$ low rank approximation (used by Song et al. (2019a); Chierichetti et al. (2017)): rank-$k$ Singular Value Decomposition (SVD).

**Datasets.** We demonstrate the benefits of our $k$-CSS$_1$ protocol on one synthetic data and one real-world application. We present a summary of the datasets, along with the number of servers $s$, the column distribution across servers and the rank $k$ we consider for each dataset in Table 1. The synthetic dataset constructs a data matrix $M \in \mathbb{R}^{(k+n) \times (k+n)}$ such that the top left $k \times k$ submatrix is the identity matrix multiplied by $n^{\frac{3}{2}}$, and the bottom right $n \times n$ submatrix has all 1's. The optimal rank-$k$ left factor consists of one of the last $n$ columns along with $k - 1$ of the first $k$ columns, incurring an error of $n^{\frac{3}{2}}$ in the $\ell_1$ norm and an error $n^3$ in the squared $\ell_2$ norm. SVD, however, will not cover any of the last $n$ columns, and thus will get an error of $n^2$ in both the $\ell_1$ and squared $\ell_2$ norms. We set $n = 2000$ and apply i.i.d. Gaussian noise to each entry with mean 0 and standard deviation 0.01. We consider a real-world application, term-document clustering, where $k$-CSS$_2$ algorithm is previously applied (Mahoney & Drineas, 2009). TechTC[3] contains 139 documents processed in bag-of-words representation with a dictionary of 18446 words. Such representation naturally results in a sparse matrix. $k$-CSS$_p$ is used to select the top $k$ most representative words.

**Hyperparameters.** Experiment hyperparameters are summarized in Table 2. We denote the number of rows in our 1-stable (Cauchy) sketching matrix by `cauchy size`, and the strong coreset size by `coreset size`. We have two additional hyperparameters for regular $k$-CSS$_{1,2}$. We denote the number of rows in the sparse embedding matrix of $\widetilde{O}(k)$ rows by `sketch size`, and the number of non-zero entries in each column of the sparse embedding matrix by `sparsity`.

---

[3]http://gabrilovich.com/resources/data/techtc/techtc300/techtc300.html

|  | synthetic | | TechTC | |
|---|---|---|---|---|
|  | *Greedy* | *Regular* | *Greedy* | *Regular* |
| cauchy size | 8k | 8k | 40 | 40 |
| coreset size | 10 | 10k | $\min(4 \times$ cauchy size$, 6k)$ | $\min(4 \times$ cauchy size$, 6k)$ |
| sketch size | - | k/3 | - | k/3 |
| sparsity | - | min(5, k/3) | - | min(5, k/3) |

Table 2: A summarization of hyperparameters used for each dataset. Note that a too small `coreset size` compared to `cauchy size` will incur large $\ell_1$ cost, and `coreset size` needs to be increased as rank $k$ increases.

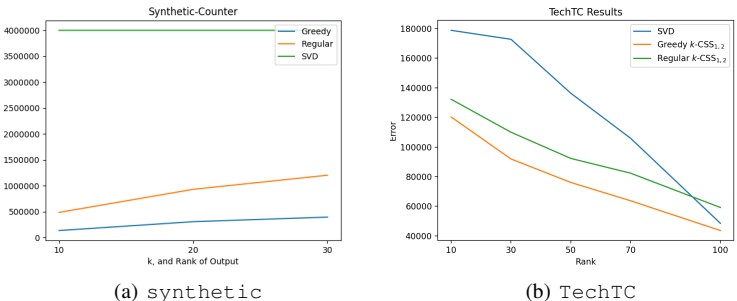

(a) `synthetic`

(b) `TechTC`

Figure 2: Results on `synthetic` and `TechTC`. The green line denotes Greedy $k$-CSS$_{1,2}$, the orange lines denotes Regular $k$-CSS$_{1,2}$, and the blue line denotes SVD.

**Setup.** We conduct 15 trials for each experiment setting, and report the average $\ell_1$ error, i.e. $|\hat{A}_k - A|_1$ for a $\widetilde{O}(k)$-rank matrix $\hat{A}_k$ output by the algorithm that approximates the original data matrix $A \in \mathbb{R}^{d \times n}$, across all trials. However, we notice that computing the $\ell_1$ error for our protocol involves computing an $\ell_1$ regression on all columns of the data matrix, which takes prohibitively long time on the real-world dataset `TechTC`. Therefore, for `TechTC`, we proceed with an approximate computation as follows: for the output subset of columns $A_I$, first i.i.d. sample 30% columns from $A$, denoted as $A_J$, with probability proportional to $|A_{*j}|_1, \forall j \in [n]$; then compute a lower bound for the $\ell_1$ error $L = \sum_{j \in J} \min_v |A_I v - A_{*j}|_1$; after that, sample $\frac{20|A|_1}{L}$ columns from $A$, denoted as $A_K$, with probability again proportional to $|A_{*j}|_1, \forall j \in [n]$; finally, the $\ell_1$ error is computed as $\frac{L}{|A|_1} \cdot \sum_{k \in K} \frac{1}{20|A_{*k}|_1} \min_v |A_I v - A_{*k}|_1$. The approximated $\ell_1$ error can be shown as an unbiased estimation of $\min_V |A_I V - A|_1$ with small variance by Hoeffding's bound.

**Results.** We present our empirical results in Figure 2. The distributed protocol performs better using GREEDY $k$-CSS$_{1,2}$ than REGULAR $k$-CSS$_{1,2}$ on both datasets, and in other settings we include in the supplementary material.

We also conducted experiments on three different datasets, `bcsstk13`, `isolet`, and `caltech-101`, extensively comparing the approximation error as well as the running time for different algorithms. A comprehensive reporting of our results is given in Appendix G.

## 7 CONCLUSION

In this work, we give the first nearly-optimal communication and number of rounds protocol for distributed $k$-CSS$_p$ $(1 \le p < 2)$ in the column partition model, which achieves $\widetilde{O}(k^{1/p-1/2})$-approximation to the best possible subset of columns, with $\widetilde{O}(sdk)$ communication cost, 1 rounds and polynomial time. To achieve a good approximation factor, we use dense $p$-stable sketching and work with the $\ell_{p,2}$ norm, which enables us to use an efficient construction of strong coresets and an $O(1)$-approximation bi-criteria $k$-CSS$_{p,2}$ algorithm. We further propose a greedy algorithm for $k$-CSS$_{1,2}$ and show the first additive error upper bound compared to the best possible subset of columns. We implement our distributed protocol using both greedy $k$-CSS$_{1,2}$ and regular $k$-CSS$_{1,2}$. Our results empirically show that greedy $k$-CSS$_{1,2}$ gives substantial improvements over regular $k$-CSS$_{1,2}$ on real-world datasets. For future works, it is not known whether a $\widetilde{O}(k^{1/p-1/2})$-approximation factor to the best possible subset of columns, is optimal for distributed $k$-CSS$_p$ $(1 \le p < 2)$.

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

## A  A HIGH COMMUNICATION COST PROTOCOL FOR $k$-CSS$_p$ ($p \geq 1$)

We start by describing a protocol for distributed $k$-CSS$_p$, which works for all $p \geq 1$, in the column partition model, and which achieves an $O(k^2)$-approximation to the best rank-$k$ approximation, using $O(1)$ rounds and polynomial time but requiring a communication cost that is linear in $n + d$. The inputs are a column-wise partitioned data matrix $A \in \mathbb{R}^{d \times n}$ distributed across $s$ servers and a rank parameter $k \in \mathbb{N}$. Each server $i$ holds part of the data matrix $A_i \in \mathbb{R}^{d \times n_i}$, $\forall i \in [s]$, and such that $\sum_{i=1}^{s} n_i = n$.

We use a single machine, polynomial time bi-criteria $k$-CSS$_p$ algorithm as a subroutine of the protocol, e.g., Algorithm 3 in Chierichetti et al. (2017), which selects a subset of $\widetilde{O}(k)$ columns $A_T$ of the data matrix $A \in \mathbb{R}^{d \times n}$ in polynomial time, for which $\min_X |A_T X - A|_p \leq O(k) \min_{\text{rank-k } A_k} |A - A_k|_p, \forall p \geq 1$.

---

**Algorithm 3** A protocol for $k$-CSS$_p$ ($p \geq 1$)

---

**Initial State:**  Server $i$ holds matrix $A_i \in \mathbb{R}^{d \times n_i}$, $\forall i \in [s]$.
**Server $i$:**
Apply polynomial time bi-criteria $k$-CSS$_p$ on $A_i$ to obtain a subset $B_i$ of columns as the left factor.
Solve for the right factor $V_i = \arg\min_{V_i} |U_i V_i - A_i|_p$. Send $U_i$ and $V_i$ to the coordinator.
**Coordinator:**
Column-wise concatenate the $U_i V_i$ to obtain $UV = [U_1 V_1, \ldots, U_s V_s]$. Apply a polynomial time bi-criteria $k$-CSS$_p$ algorithm on $UV$ to obtain a subset $C$ of columns. Send $C$ to each server.
**Server $i$:**
Solve $\min_{X_i} |C X_i - A_i|_p$ to obtain the right factor.

---

**Approximation Factor.**  Let $UV$ denote the column-wise concatenation of the $U_i V_i$. Let $X^* = \arg\min_X |CX - A|_p$. Then,

$$
\begin{aligned}
|CX^* - AS|_p &\leq |CX^* - UV|_p + |UV - A|_p && \text{By the triangle inequality} \\
&\leq O(k) \min_{\text{rank-k } (UV)_k} |UV - (UV)_k|_p + |UV - A|_p && \text{By the } O(k)\text{-approximation of } k\text{-CSS}_p \\
&\leq O(k)|UV - A|_p \\
&= O(k)(\sum_{i=1}^{s} |U_i V_i - A_i|_p) \\
&\leq O(k)(\sum_{i=1}^{s} O(k) \min_{\text{rank-k } A_i^*} |A_i - A_i^*|_p) && \text{By the } O(k)\text{-approximation of } k\text{-CSS}_p \\
&\leq O(k^2) \sum_{i=1}^{s} |A_i - (A^*)_i|_p && A^* = \arg\min_{\text{rank-k } A^*} |A - A^*|_p \\
&= O(k^2)|A - A^*|_p
\end{aligned}
$$

**Communication Cost.**  Since $U_i \in \mathbb{R}^{d \times \widetilde{O}(k)}$ and $V_i \in \mathbb{R}^{\widetilde{O}(k) \times n_i}$, sending $U_i$ and $V_i$ costs $\widetilde{O}(skn)$. Since $C \in \mathbb{R}^{d \times \widetilde{O}(k)}$, sending $C$ from the coordinator to all servers costs $\widetilde{O}(sdk)$. Thus the overall communication cost is $\widetilde{O}(s(n + d)k)$.

**Running time.**  According to Chierichetti et al. (2017), applying the $k$-CSS$_p$ algorithm and solving $\ell_p$ regression can both be done in polynomial time. Thus the overall running time of the protocol is polynomial.

**Problems with this protocol.**  Although this protocol works for all $p \geq 1$, a communication cost that linearly depends on the large dimension $n$ is too high, and furthermore, the output $C$ is not a subset of columns of $A$, because the protocol applies $k$-CSS$_p$ on a concatenation of both the left factor $U_i$ and the right factor $V_i$. $U_i$ is a subset of columns of $A_i$ but $V_i$ is not necessarily a sampling matrix. One might wonder whether it is possible that each server only sends $U_i$ and the coordinator then runs $k$-CSS$_p$ on a concatenation of the $U_i$. This will not necessarily give a good approximation to $\min_{\text{rank-k } A_k} |A - A_k|_p$ because the columns not selected in the $U_i$ locally on each server might become globally important. Finally, although it is possible to improve the approximation factor to $\widetilde{O}(k)$ by making use of an $\widetilde{O}(\sqrt{k})$-approximation algorithm for $\ell_p$-low rank approximation that also selects a subset of columns (Mahankali & Woodruff, 2020), this protocol would still suffer from all of the aforementioned problems.

## B   Preliminaries for Our Upper Bound Proofs

### B.1   Norms

**Lemma 1.** *(Norm Relationships) For a matrix $A \in \mathbb{R}^{d \times n}$, $|A|_{p,2} \leq |A|_p$ and $|A|_p \leq d^{\frac{1}{p} - \frac{1}{2}} |A|_{p,2}$, where $1 \leq p < 2$.*

*Proof.* Let $x \in \mathbb{R}^d$. For $0 < p < r$,

$$|x|_r \leq |x|_p \leq d^{\frac{1}{p} - \frac{1}{r}} |x|_r$$

Let $r = 2$. Then we have

$$|x|_2 \leq |x|_p \leq d^{\frac{1}{p} - \frac{1}{2}} |x|_2$$

Note that $|A|_{p,2} = \left( \sum_j |A_{*j}|_2^p \right)^{\frac{1}{p}}$ and $|A|_p = \left( \sum_j |A_{*j}|_p^p \right)^{\frac{1}{p}}$.

Therefore,

$$|A|_{p,2} = \left( \sum_j |A_{*j}|_2^p \right)^{\frac{1}{p}} \leq \left( \sum_j |A_{*j}|_p^p \right)^{\frac{1}{p}} = |A|_p$$

and

$$|A|_p = \left( \sum_j |A_{*j}|_p^p \right)^{\frac{1}{p}} \leq d^{\frac{1}{p} - \frac{1}{2}} \left( \sum_j |A_{*j}|_2^p \right)^{\frac{1}{p}} = d^{\frac{1}{p} - \frac{1}{2}} |A|_{p,2}$$

$\square$

### B.2   Lower Bound on the Cost - No Contraction when Applying a $p$-stable Sketch

We show a lower bound on the approximation error for a sketched subset of columns, $|SA_T V - SA|_p$, in terms of $|A_T V - A|_p$. The lower bound holds simultaneously for any arbitrary subset $A_T$ of chosen columns, and for any arbitrary right factor $V$.

We begin the proof by first showing that applying a dense $p$-stable sketch to a vector will not shrink its $p$-norm. This is done in **Lemma** 2.1. We further observe that although $p$-stable random variables are heavy-tailed, we can still bound their tail probabilities by applying Lemma 9 from Meng & Mahoney (2013). We note this in **Lemma** 2.2. Note that the $X_i$'s do not need to be independent in this lemma.

Equipped with **Lemma** 2.1, **Lemma** 2.2 and a net argument, we can now establish a lower bound on $|SA_T V - SA|_p$. We first show in **Lemma** 2.3 that, with high probability, for any arbitrarily selected subset $A_T$ of columns and for an arbitrary column $A_{*j}$, the error incurred to fit $SA_{*j}$ using the columns of $SA_T$ is no less than the error incurred to fit $A_{*j}$ using the columns of $A_T$. We then apply a union bound over all subsets $T \subset [n]$ and columns $j \in [n]$ to conclude our lower bound in **Lemma** 2.

**Lemma 2.1.** *(No Contraction of p-stable Sketch) Given a matrix $S \in \mathbb{R}^{t \times m}$ whose entries are i.i.d. p-stable random variables rescaled by $\Theta \left( \frac{1}{t^{\frac{1}{p}}} \right)$, where $1 \leq p < 2$, for any fixed $y \in \mathbb{R}^m$, with probability $1 - \frac{1}{e^t}$, the following holds:*

$$|Sy|_p \geq |y|_p$$

*Proof.* By $p$-stability, we have $|Sy|_p^p = \sum_{i=1}^{t} \left( |y|_p \frac{|Z_i|}{t^{\frac{1}{p}}} \right)^p$, where $|Z_i| \geq 0$ are half $p$-stable random variables. Since $Pr[|Z_i| = \Omega(1)] > \frac{1}{2}$, by applying a Chernoff bound (to the indicators $1_{|Z_i| \geq C}$ for a sufficiently small constant $C$), we have $\sum_{i=1}^{t} |Z_i|^p = \Omega(t)$ with probability $1 - \frac{1}{e^t}$. Therefore, with probability $1 - \frac{1}{e^t}$, $|Sy|_p \geq |y|_p$. $\square$

**Lemma 2.2.** *(Upper Tail Inequality for p-stable Distributions) Let $p \in (1, 2)$, and $m > 3$. For $i \in [m]$, let $X_i$ be a standard p-stable random variable, and let $\gamma_i > 0$ and $\gamma = \sum_{i=1}^{m} \gamma_i$. Let $X = \sum_{i=1}^{m} \gamma_i |X_i|^p$. Then, for any $t \geq 1$, $Pr[X \geq t\alpha_p \gamma] \leq \frac{2 \log(mt)}{t}$, where $\alpha_p > 0$ is a constant that is at most $2^{p-1}$.*

*Proof.* Lemma 9 from Meng & Mahoney (2013) for $p \in (1, 2)$. $\square$

**Lemma 2.3.** *(No Contraction for All Sketched Subsets and Columns) Let $A \in \mathbb{R}^{d \times n}$, and $k \in \mathbb{N}$. Let $t = k \cdot poly(\log nd)$, and let $S \in \mathbb{R}^{t \times d}$ be a matrix whose entries are i.i.d. standard p-stable random variables, rescaled by $\Theta(1/t^{\frac{1}{p}})$. Finally, let $m = k \cdot poly(\log k)$. Then, with probability $1 - o(1)$, for all $T \subset [n]$ with $|T| = m$, for all $j \in [n]$, and for all $y \in \mathbb{R}^{|T|}$,*

$$|A_T y - A_{*j}|_p \leq |S(A_T y - A_{*j})|_p$$

*Proof.* **Step 1:** We first extend **Lemma** 2.1 and argue that applying a $p$-stable sketching matrix $S \in \mathbb{R}^{t \times n}$ will not shrink the norm $|Sy|_p \geq |y|_p$ $(1 \leq p < 2)$ simultaneously for *all* $y$ in the column span of $[A_T, A_j] =: A_{T,j}$, by a net argument.

In order to bound the $p$-norm of sketched vectors $y$ in a net, we begin by showing that with high probability all entries in $S$ are bounded. Let $D = poly(mt)$. Consider the following two cases:

**Case 1:** $p = 1$. The entries of the 1-stable sketching matrix $S_{ij}$ are standard Cauchy random variables. Consider half-Cauchy random variables $X_{i,j} = |S_{i,j}|$. The cumulative distribution function of half Cauchy random variables $x$ is $F(x) = \int_0^x \frac{2}{\pi(t^2+1)} dt = 1 - \Theta(\frac{1}{x})$. Thus, for any $i \in [t]$ and $j \in [m]$, $Pr[|S_{ij}| \leq D] = 1 - \Theta(\frac{1}{D})$.

**Case 2:** $p \in (1, 2)$. We apply the upper tail bound of $p$-stable random variables in **Lemma** 2.2. For any fixed $i \in [t]$ and $j \in [m]$, $Pr[|S_{ij}|^p \leq D^p] \geq 1 - \Theta(\frac{\log(t)}{D^p})$, which implies $Pr[|S_{ij}| \leq D] \geq 1 - \Theta(\frac{1}{D})$.

Therefore, for $p \in [1, 2)$, by a union bound over all entries in $S$, if we define the event $\mathcal{E}_1$ to mean that for all $i \in [t]$ and $j \in [m]$, we simultaneously have $|S_{ij}| \leq D$, then $Pr[\mathcal{E}_1] \geq 1 - \Theta(\frac{mt}{D})$ by a union bound. The event $\mathcal{E}_1$ occurring implies that for any $y \in \mathbb{R}^d$, since all entries in $S$ are rescaled by $O(1/t^{1/p})$,

$$|Sy|_p \leq |y|_p t^{1/p} |S|_\infty \leq D|y|_p$$

Consider the unit $\ell_p$ ball $B = \{y \in \mathbb{R}^d : |y|_p = 1, \exists z \in \mathbb{R}^m \text{ s.t. } y = A_{T,j}z\}$ in the column span of $A_{T,j}$. A subset $\mathcal{N} \subset B$ is a $\gamma$-net for $B$ if for all $y \in B$ there exists some $u \in \mathcal{N}$ such that $|y - u|_p \leq \gamma$, for some distance $\gamma > 0$. There exists such a net $\mathcal{N}$ for $B$ of size $|\mathcal{N}| = (\frac{1}{\gamma})^{O(m)}$ by a standard greedy construction, since the column span of $A_{T,j}$ has dimension at most $m + 1$. We choose $\gamma = \frac{1}{m^2 D}$, and thus $|\mathcal{N}| \leq (m^2 D)^{O(m)} = D^{O(m \log m)}$.

By applying **Lemma** 2.1, and a union bound over all vectors $y \in \mathcal{N}$, we have that event $\mathcal{E}_2$: for all $y \in \mathcal{N}$ simultaneously, $|Sy|_p \geq |y|_p$ — $\mathcal{E}_2$ has probability at least $1 - \frac{D^{O(m \log m)}}{e^t}$.

Consider an arbitrary unit vector $x \in B$. There exists some $y \in \mathcal{N}$ such that $|x - y|_p \leq \gamma = \frac{1}{m^2 D}$. Conditioning on both events $\mathcal{E}_1$ and event $\mathcal{E}_2$, we have the following with probability $1 - \Theta(\frac{mt}{D}) - \frac{D^{O(m \log m)}}{e^t}$:

$$
\begin{aligned}
|Sx|_p &\geq |Sy|_p - |S(x - y)|_p && \text{Triangle Inequality} \\
&\geq |y|_p - |S(x - y)|_p && \text{By event } \mathcal{E}_2 \\
&\geq |y|_p - D|(x - y)|_p && \text{Implication of event } \mathcal{E}_1 \\
&\geq |y|_p - D\gamma && \text{By } |x - y|_p \leq \gamma \\
&= |y|_p - O(\frac{1}{m^2}) \\
&= |x|_p - O(\frac{1}{m^2}) && |x|_p = |y|_p = 1
\end{aligned}
$$

For a sufficiently large $m$, $O(\frac{1}{m^2})$ is at most $\frac{1}{2}$, and thus $\frac{|x|_p}{2} = \frac{1}{2} \geq \frac{1}{m^2}$. This implies $|Sx|_p \geq |x|_p - \frac{|x|_p}{2} = \frac{|x|_p}{2}$. We can rescale $S$ by a factor of 2 so that $|Sx|_p \geq |x|_p$.

We have shown that $|Sy|_p \geq |y|_p$ holds simultaneously for *all* unit vectors $y$ in the column span of $A_{T,j}$. By linearity, we conclude that $|Sy|_p \geq |y|_p$ $(1 \leq p < 2)$ holds simultaneously for *all* $y$ in the column span of $A_{T,j}$.

**Step 2:** Next, we apply a union bound over all possible subsets $T \subset [n]$ of chosen columns from $A$ and all possible single columns $A_{*j}$ for $j \in [n]$, to argue that $|S(A_T y - A_{*j})|_p \geq |A_T y - A_{*j}|_p$ holds for all $y \in \mathbb{R}^{|T|}$ and all $T \subset [n], j \subset [n]$ with high probability. Note that $|T| = m = O(k \cdot poly(\log k))$.

From **Step 1**, we have shown that event $\mathcal{E}_2$ fails with probability $\frac{D^{O(m \log m)}}{e^t}$. Thus event $\mathcal{E}_2$ fails over all possible subsets $T$ and all possible single columns $A_{*j}$ with probability at most

$$\frac{D^{O(m \log m)}}{e^t} \cdot \binom{d}{m-1} \cdot d \leq \frac{D^{O(m \log m)}}{e^t} \cdot d^{O(m)} \cdot d$$

The above failure probability is $o(1)$ as long as

$$
\begin{aligned}
t &= \Theta\Big( \log\Big( D^{O(m \log m)} \cdot d^{O(m)} \cdot d) \big) \Big) \\
&= \Theta(O(m \log m) \log(D) + O(m) \log d) \\
&= \Theta(O(k\mathrm{poly}(\log k)) \log(mt) + O(k\mathrm{poly}(\log k) \log(d)))
\end{aligned}
$$

Thus it suffices to have $t = k\mathrm{poly}(\log nd)$ to have failure probability at most $\frac{1}{e^{O(t)}}$.

Now we condition on a single global event $\mathcal{E}_1$ and that event $\mathcal{E}_2$ holds for all possible $T$ and $A_{*j}$. We conclude that with probability $1 - \Theta(\frac{mt}{D}) - \frac{1}{e^{O(t)}}$, the following holds simultaneously for all $T \subset [n]$ and for all $A_{*j}$ for which $j \in [n]$:

$$
|A_T y - A_{*j}|_p \leq |S(A_T y - A_{*j})|_p.
$$

$\square$

**Lemma 2.** *(Lower Bound for Sketched Error) Let $A \in \mathbb{R}^{d \times n}$ and $k \in \mathbb{N}$. Let $t = k \cdot poly(\log(nd))$, and let $S \in \mathbb{R}^{t \times d}$ be a matrix whose entries are i.i.d. standard p-stable random variables, rescaled by $\Theta(1/t^{\frac{1}{p}})$. Then, with probability $1 - o(1)$, for all $T \subset [n]$ with $|T| = k \cdot poly(\log k)$ and for all $V \in \mathbb{R}^{|T| \times n}$,*

$$
|A_T V - A|_p \leq |SA_T V - SA|_p
$$

*Proof.* Let $y_j$ denote the $j$-th column of $V$, where $j \in [n]$. By applying **Lemma** 2.3, and a union bound over all columns of $V$, for $m = |T| = k \cdot \mathrm{poly}(\log k)$, $t = k \cdot \mathrm{poly}(\log(nd))$ and $D = \mathrm{poly}(mt) = \mathrm{poly}(k \log(nd))$, the following holds with probability $1 - \Theta(\frac{mt}{D}) - \frac{n}{e^{O(t)}} = 1 - o(1)$,

$$
\begin{aligned}
|A_T V - A|_p &= \Big( \sum_{j=1}^n |A_T y_j - A_j|_p^p \Big)^{\frac{1}{p}} \\
&\leq \Big( \sum_{j=1}^n |S(A_T y_j - A_j)|_p^p \Big)^{\frac{1}{p}} \\
&= |SA_T V - SA|_p
\end{aligned}
$$

$\square$

### B.3 UPPER BOUND ON THE COST

We show an upper bound on the approximation error of $k$-CSS$_p$ on a sketched subset of columns, $|SA_T V - SA_T|_p$, which holds for a fixed subset $A_T$ of columns and for the minimizing right factor $V = \arg\min_V |SA_T V - SA|_p$ for that subset of columns.

We first adapt Lemma E.17 from Song et al. (2017) to establish an upper bound on the error $|SA_T V - SA|_p$ for any fixed $V$ in **Lemma** 3.1. We then apply **Lemma** 3.1 to the minimizer $V$ to conclude the upper bound in **Lemma** 3.

**Lemma 3.1.** *(An Upper Bound on Norm of A Sketched Matrix) Given $A \in \mathbb{R}^{n \times d}$ and $p \in [1, 2)$, and $U \in \mathbb{R}^{n \times k}$ and $V \in \mathbb{R}^{k \times d}$, if $S \in \mathbb{R}^{t \times n}$ is a dense p-stable matrix, whose entries are rescaled by $\Theta\left( \frac{1}{t^{\frac{1}{p}}} \right)$, then with probability at least $1 - O(1)$,*

$$
|SUV - SA|_p^p \leq O(\log(td))|UV - A|_p^p
$$

*Here, the failure probability $O(1)$ can be arbitrarily small.*

*Proof.* Lemma E.17 from Song et al. (2017). $\square$

**Lemma 3.** *(Upper Bound on Sketched Error) Let $A \in \mathbb{R}^{d \times n}$ and $k \in \mathbb{N}$. Let $t = k \cdot poly(\log(nd))$, and let $S \in \mathbb{R}^{t \times d}$ be a matrix whose entries are i.i.d. standard p-stable random variables, rescaled by $\Theta(1/t^{\frac{1}{p}})$. Then, for a fixed subset $T \subset [n]$ of columns with $|T| = k \cdot poly(\log k)$, with probability $1 - O(1)$, we have*

$$
\min_V |SA_T V - SA|_p \leq \min_V O(\log^{1/p}(nd))|A_T V - A|_p
$$

*Here, the failure probability can be an arbitrarily small constant.*

*Proof.* Let $X_1^* = \arg\min_X |SA_T X - SA|_p$ and $X_2^* = \arg\min_X |A_T X - A|_p$. By **Lemma** 3.1,

$$|SA_T X_1^* - SA|_p^p \le |SA_T X_2^* - SA|_p^p$$
$$\le O(\log(k\mathrm{poly}(\log n)d))|A_T X_2^* - A|_p^p$$
$$\le O(\log(nd))|A_T X_2^* - A|_p^p$$

Therefore,

$$\min_X |SA_T X - SA|_p \le \min_X O(\log^{1/p}(nd))|A_T X - A|_p$$

. $\qquad\qquad\qquad\qquad\qquad\qquad\qquad\qquad\qquad\qquad\qquad\qquad\qquad\qquad\qquad\qquad\square$

### B.4 Strong Coresets in the $\ell_{p,2}$ Norm

**Lemma 4** (Strong Coreset in $\ell_{p,2}$ norm). *Let $A \in \mathbb{R}^{d \times n}$, $k \in \mathbb{N}$, $p \in [1,2)$, and $\epsilon, \delta \in (0,1)$. Then, in $n \cdot poly(k \log n/\epsilon)$ time, one can find a sampling and reweighting matrix $T$ with $O(d \log d/\epsilon^2) \cdot \log(1/\delta)$ columns such that, with probability $1 - \delta$, for all rank-k matrices $U$,*

$$\min_{rank\text{-}k\ V} |UV - AT|_{p,2} = (1 \pm \epsilon) \min_{rank\text{-}k\ V} |UV - A|_{p,2}$$

*AT is called a **strong coreset** of A.*

*Proof.* We can obtain $T$ with $O(d(\log d)/\epsilon^2)$ columns using the strong coreset construction from **Lemma 16** in Sohler & Woodruff (2018). Note that the coreset construction for $k$-subspace approximation in Sohler & Woodruff (2018) aims at removing a dependence on $d$ in the coreset size. The algorithm first finds a poly$(k)$-dimensional subspace $S$ by running a dimensionality reduction algorithm and constructs coresets in the lower dimensional subspace $S$, resulting in a coreset size of poly$(k/\epsilon)$. But in our case, we do not want our coreset size to have a polynomial dependency on $k$ while a linear dependency on $d$ suffices. Thus, instead of running their dimensionality reduction algorithm to find such a subspace $S$ to project $A$ to, we directly use the the column span of the input matrix $A$ as the subspace $S$ appended with a row of zeros to construct the $B$ in **Lemma 16** of Sohler & Woodruff (2018). Note that in the original algorithm, the appended column encodes the distances from the input matrix $A$ to the subspace $S$, but in our case it is just all 0's. Then the guarantees and running time claimed above immediately follow from **Lemma 16** of Sohler & Woodruff (2018).

We note that the size of our coreset for $k$-subspace approximation can be further reduced to $\widetilde{O}(k)$, suppressing a logarithmic dependence on $k, \frac{1}{\epsilon}, \frac{1}{\delta}$, using an additional $O(nd)$ time, by combining **Corollary 5.16** of Huang & Vishnoi (2020) and the importance sampling scheme from Stage 2 of Algorithm 1 in Huang & Vishnoi (2020). Furthermore, though not explicitly stated in **Lemma 16** of Sohler & Woodruff (2018), the coreset size has a $\log(\frac{1}{\delta})$ dependence on the failure probability $\delta$ due to the importance sampling (Sohler & Woodruff, 2018; Huang & Vishnoi, 2020). $\qquad\square$

Note that an accuracy of $(1 \pm \epsilon)$ is desired for our extension to the streaming model, described in Appendix E, and for both Algorithm 1 and Algorithm 7, we perform a union bound over strong coreset constructions, for which a $\log(1/\delta)$ dependence on $\delta$ is sufficient.

## C Polynomial Time, $O(1)$-approximate Bi-criteria $k$-CSS$_{p,2}$

We give a detailed analysis on the polynomial time, $O(1)$-approximate $k$-CSS$_{p,2}$ algorithm presented in Algorithm 4, which is based on ideas in Clarkson & Woodruff (2015). We first use a sparse embedding matrix $S$ to obtain an $O(1)$-approximate left factor. We then use $\ell_p$-Lewis weight sampling Cohen & Peng (2015) to select a subset of columns.

---

**Algorithm 4** polynomial time, $O(1)$-approximation for $k$-CSS$_{p,2}$ ($1 \le p < 2$)

---

**Input:** The data matrix $A \in \mathbb{R}^{d \times n}$, rank $k \in \mathbb{N}$
**Output:** The left factor $U \in \mathbb{R}^{d \times \widetilde{O}(k)}$, the right factor $V \in \mathbb{R}^{\widetilde{O}(k) \times n}$ such that $|UV - A|_{p,2} \le O(1)\min_{\text{rank-k } A_k} |A_k - A|_{p,2}$
$S \leftarrow \widetilde{O}(k) \times d$ sparse embedding matrix, with sparsity $s = \mathrm{poly}(\log k)$.
$S' \leftarrow n \times \widetilde{O}(k)$ sampling matrix, each column of which is a standard basis vector chosen randomly according to the $\ell_p$ Lewis weights of columns of $SA$.
**return** $U \leftarrow AS'$, $V \leftarrow (AS')^\dagger A$ {$\dagger$ denotes the Moore-Penrose pseudoinverse.}

---

## C.1 SPARSE EMBEDDING MATRICES

The **sparse embedding matrix** $S \in \mathbb{R}^{\widetilde{O}(k) \times d}$ of Nelson & Nguyen (2013), and used by Clarkson & Woodruff (2015), is constructed as follows: each column of $S$ has exactly $s$ non-zero entries chosen in uniformly random locations. Each non-zero entry is a random value $\pm \frac{1}{\sqrt{s}}$ with equal probability. $s$ is also called the *sparsity* of $S$. Let $h$ be the hash function that picks the location of the non-zero entries in each column of $S$ and $\sigma$ be the hash function that determines the sign $\pm$ of each non-zero entry.

Applying the sparse embedding matrix $S$ to $A$ enables us to obtain a rank-$k$ right factor that is at most a factor of $O(1)$ worse than the best rank-$k$ approximation error in the $\ell_{p,2}$ norm. We adapt Theorem 32 from Clarkson & Woodruff (2015) to show this in **Theorem** 5.5. Notice that in Theorem 32 of Clarkson & Woodruff (2015), the number of rows required for $S$ is $O(k^2)$, but this can be reduced to $\widetilde{O}(k)$ through a different choice of hyperparameters when constructing the sparse embedding matrix $S$.

We note two choices of hyperparameters, i.e., the number $m$ of rows and sparsity $s$, of $S$ in **Theorem** 5.1 and **Theorem** 5.2, both of which give the same result. The proof of Theorem 32 from Clarkson & Woodruff (2015) uses the hyperparameters from **Theorem** 5.1. We replace the hyperparameters from **Theorem** 5.2 and show in **Lemma** 5.3 that $\widetilde{O}(k)$ rows of $S$ suffice to preserve certain desired properties. We then combine **Lemma** 5.3 and **Lemma** 5.4 adapted from Clarkson & Woodruff (2015), to conclude our result in **Theorem** 5.5, following the analysis from Clarkson & Woodruff (2015).

**Theorem 5.1.** *(Theorem 3 from Nelson & Nguyen (2013)) For a sparse embedding matrix $S \in \mathbb{R}^{m \times n}$ with sparsity $s = 1$ and a data matrix $U \in \mathbb{R}^{n \times d}$, let $\epsilon \in (0,1)$. With probability at least $1 - \delta$ all singular values of $SU$ are $(1 \pm \epsilon)$ as long as $m \geq \delta^{-1}(d^2 + d)/(2\epsilon - \epsilon^2)^2$. For the hash functions used to construct $S$, $\sigma$ is 4-wise independent and $h$ is pairwise independent.*

**Theorem 5.2.** *(Theorem 9 from Nelson & Nguyen (2013)) For a sparse embedding matrix $S \in \mathbb{R}^{m \times n}$ with sparsity $s = \Theta(\log^3(d/\delta)/\epsilon)$ and a data matrix $U \in \mathbb{R}^{n \times d}$, let $\epsilon \in (0,1)$. With probability at least $1 - \delta$ all singular values of $SU$ are $(1 \pm \epsilon)$ as long as $m = \Omega(d \log^8(d/\delta)/\epsilon^2)$. For the hash functions used to construct $S$, we have that $\sigma, h$ are both $\Omega(\log(d/\delta))$-wise independent.*

**Lemma 5.3.** *Let $\mathcal{C}$ be a constraint set and $A \in \mathbb{R}^{n \times d}, B \in \mathbb{R}^{n \times d'}$ be two arbitrary matrices. For a sparse embedding matrix $S \in \mathbb{R}^{m \times n}$, there is $m = O(\frac{d \log^8(\frac{d}{\epsilon^{p+1}})}{\epsilon^{2(p+1)}})$, such that with constant probability, the following holds for all $X \in \mathbb{R}^{d \times d'}$*

*i)* $|S(AX - B)|_{p,2} \geq (1-\epsilon)|AX - B|_{p,2}$

*ii)* $|S(AX^* - B)|_{p,2} \leq (1+\epsilon)|AX^* - B|_{p,2}$, *where* $X^* = \underset{X \in \mathcal{C}}{\arg\min} |AX - B|_{p,2}$

*Proof.* The proof is the same as the proof of Lemma 29 from Clarkson & Woodruff (2015), except that we use a different choice of hyperparameters in constructing $S$, i.e., sparsity $s$ and the number $m$ of rows. In the proof of Lemma 29 from Clarkson & Woodruff (2015), the construction of $S$ follows **Theorem** 5.1, where the sparsity $s = 1$, but requires $m = O(d^2)$ rows. We replace the construction by **Theorem** 5.2, where we pick $\delta = \epsilon^{p+1}$. Now the sparsity $s$ is larger but this construction reduces the number of rows required to $m = \widetilde{O}(d)$. Since both choices of hyperparameters to construct $S$ result in bounded $(1 \pm \epsilon)$ singular values of $SU$ for any data matrix $U$, the rest of the proof follows. □

**Lemma 5.4.** *Consider a data matrix $A \in \mathbb{R}^{n \times d}$. Let the best rank-$k$ matrix in the $\ell_{p,2}$ norm be $A_k = \arg\min_{rank\text{-}k\ A_k} |A_k - A|_{p,2}$. For $R \in \mathbb{R}^{d \times m}$, if $R^T$ satisfies both of the following two conditions for all $X \in \mathbb{R}^{n \times n}$:*

*i)* $|R^T(A_k^T X - A^T)|_{p,2} \geq (1-\epsilon)|A_k^T X - A^T|_{p,2}$

*ii)* $|R^T(A_k^T X^* - A^T)|_{p,2} \leq (1+\epsilon)|A_k^T X^* - A^T|_{p,2}$, *where* $X^* = \underset{X}{\arg\min} |A_k^T X - A^T|_{p,2}$

*then*

$$\min_{rank\text{-}k\ X} |XR^T A_k^T - A^T|_{p,2}^p \leq (1+3\epsilon)|A_k^T - A^T|_{p,2}^p$$

*Proof.* Lemma 31 from Clarkson & Woodruff (2015). □

**Theorem 5.5.** *($\ell_{p,2}$-Low Rank Approximation) Let the data matrix be $A \in \mathbb{R}^{d \times n}$ and $k \in \mathbb{N}$ be the desired rank. Let $S \in \mathbb{R}^{m \times d}$ be a sparse embedding matrix with $m = O(k\,poly(\log k)poly(\frac{1}{\epsilon}))$ rows, and sparsity $s = poly(\log k)$. Then, the following holds with constant probability,*

$$\min_{rank\text{-}k\ X} |XSA - A|_{p,2} \leq (1+3\epsilon) \min_{rank\text{-}k\ A_k} |A_k - A|_{p,2}$$

*Proof.* The proof is the same as the proof of Theorem 32 in Clarkson & Woodruff (2015), except that we adapt a different construction of the sparse embedding matrix $S$, which reduces the number of rows from $O(k^2)$ to $\widetilde{O}(k)$ with increased sparsity $s$.

Consider $A_k = \arg\min_{\text{rank-k } A_k} |A_k - A|_{p,2}$. Let $V_k$ be a basis for the column space of $A_k$. By applying **Lemma** 5.3 and **Lemma** 5.4 on the basis $V_k$, we conclude the above theorem by setting the number $m$ of rows to $m = O\left(\frac{k \log^8(\frac{k}{\epsilon^2})}{\epsilon^4}\right)$, and sparsity $s = \text{poly}(\log k)$ in the sparse embedding matrix $S$.

$\square$

### C.2 LEWIS WEIGHTS SAMPLING

The $\ell_p$ **Lewis weight** is an inherent property of a matrix. By Cohen & Peng (2015), the unique set of $\ell_p$ Lewis weights $w$ for a matrix is defined as follows: for the $i$-th column of a matrix $M \in \mathbb{R}^{n \times d}$, $M_{*j}$, is defined as $w_i^{2/p} = M_{*j}^T (MW^{1-2/p}M^T)^{-1}M_{*j}$, where $W$ is the diagonal matrix with $W_{ii} = w_i, \forall i \in [d]$.

We first note that sampling by Lewis weights provides a subspace embedding in the $\ell_p$-norm in **Theorem** 5.6. We further apply a randomized version of Dvoretzky's Theorem in **Theorem** 5.7, which allows the embedding from the $\ell_p$ norm into a low-dimensional Euclidean subspace with very small distortion and thus enables us to switch between the $\ell_p$ norm and the $\ell_{p,2}$ norm. Based on **Theorem** 5.6 and **Theorem** 5.7, we show in **Theorem** 5.8 that Lewis weights sampling provides a good subset of columns, on which the analysis of $k$-CSS$_{p,2}$ is based.

**Theorem 5.6.** *($\ell_p$-Lewis Weights Subspace Embedding) Let $A \in \mathbb{R}^{n \times d}$ and $t = \widetilde{O}(d)$. For $1 \leq p < 2$, there exists a distribution $(\lambda_1, \lambda_2, \ldots, \lambda_n)$ on the rows of $A$ such that if we generate a matrix $S$ with $t$ rows, each chosen independently as the $i^{th}$ standard basis vector times $\frac{1}{(r\lambda_i)^{\frac{1}{p}}}$ with probability $\lambda_i$, then with probability $1 - o(1)$, the following holds for all $x \in \mathbb{R}^d$,*

$$\Omega(1)|Ax|_p \leq |SAx|_p \leq O(1)|Ax|_p$$

*Proof.* Theorem 7.1 from Cohen & Peng (2015)

$\square$

**Theorem 5.7.** *(Randomized Dvoretzky's Theorem) Let $n \in \mathbb{N}$, and $\epsilon \in (0, 1)$. Let $r = \frac{n}{\epsilon^2}$. Let $G \in \mathbb{R}^{r \times n}$ be a random matrix whose entries are i.i.d. standard Gaussian random variables, rescaled by $\frac{1}{\sqrt{r}}$. For $r = \frac{n}{\epsilon^2}$, the following holds with probability $1 - e^{-\Theta(n)}$, for all $y \in \mathbb{R}^n$,*

$$|Gy|_p = (1 \pm \epsilon)|y|_2$$

*Proof.* This follows from Theorem 1.2 from Paouris et al. (2017).

$\square$

**Theorem 5.8.** *(Subset of Columns by Lewis Weights Sampling) Let $A \in \mathbb{R}^{d \times n}$. Let $S \in \mathbb{R}^{m \times d}$ be a sparse embedding matrix, with $m = O(k \cdot poly(\log k)poly(\frac{1}{\epsilon}))$. Further, let $S' \in \mathbb{R}^{n \times t}$ be a sampling matrix whose columns are random standard basis vectors generated according to the $\ell_p$ Lewis weights of columns of $SA$, with $t = k \cdot poly(\log k)$. Then, for $\hat{X} = \arg\min_{\text{rank-k } X} |XSAS' - AS'|_{p,2}$, the following holds with constant probability,*

$$|\hat{X}SA - A|_{p,2} \leq \Theta(1) \min_{\text{rank-k } A_k} |A_k - A|_{p,2}$$

*Proof.* Let $X^* = \arg\min_{\text{rank-k } X^*} |X^*SA - A|_{p,2}$. By the triangle inequality,

$$|\hat{X}SA - A|_{p,2} \leq |X^*SA - \hat{X}SA|_{p,2} + |X^*SA - A|_{p,2}$$

Let us consider bounding $|X^*SA - \hat{X}SA|_{p,2}$.

By Lemma D.28 and Lemma D.29 from Song et al. (2017), for any column sampling matrix $S$ and for any fixed matrix $Y$, it can be shown that $\mathbf{E}[|YS|_p^p] = |YS|_p^p$. In our case, since $S'$ is a sampling matrix, we have $\mathbf{E}[|YS'|_p^p] = |YS'|_p^p$ for any fixed matrix $Y$.

Now let $G \in \mathbb{R}^{\Theta(d) \times d}$ be a rescaled random matrix whose entries are i.i.d. standard Gaussian random variables as in **Theorem** 5.7. We apply **Theorem** 5.7 to transform between the $\ell_p$ space and the Euclidean space. Since transformation of both directions can be done with very small distortion, we obtain a $\Theta(1)$ approximation. With constant probability, we have

$$|X^*SA - \hat{X}SA|_{p,2} = \Theta(1)|G(X^* - \hat{X})SA|_p \qquad \text{By **Theorem** 5.7}$$

$$
\begin{aligned}
&= \Theta(1)|G(X^* - \hat{X})SAS'|_p && \text{By } \textbf{Theorem } 5.6\\
&= \Theta(1)|(X^* - \hat{X})SAS'|_{p,2} && \text{By } \textbf{Theorem } 5.7\\
&\leq \Theta(1)\Big(|X^*SAS' - AS'|_{p,2} + |\hat{X}SAS' - AS'|_{p,2}\Big) && \text{Triangle Inequality}\\
&\leq \Theta(1)|X^*SAS' - AS'|_{p,2} && \text{Since } \hat{X} = \arg\min_{\text{rank-k } X} |XSAS' - AS'|_{p,2}\\
&= \Theta(1)|G(X^*SA - A)S'|_p && \text{By } \textbf{Theorem } 5.7\\
&\leq \Theta(1)|G(X^*SA - A)|_p && \text{By Markov Bound on } \mathbf{E}[|YS'|_p^p] = |YS'|\\
&= \Theta(1)|X^*SA - A|_{p,2} && \text{By } \textbf{Theorem } 5.7
\end{aligned}
$$

Therefore,

$$
\begin{aligned}
|\hat{X}SA - A|_{p,2} &\leq |X^*SA - \hat{X}SA|_{p,2} + |X^*SA - A|_{p,2}\\
&\leq \Theta(1)|X^*SA - A|_{p,2}\\
&\leq \Theta(1)\min_{\text{rank-k } A_k}|A - A_k|_{p,2} && \text{By } \textbf{Theorem } 5.5
\end{aligned}
$$

$\square$

## C.3 ANALYSIS FOR $k$-CSS$_{p,2}$

We are now ready to show an $O(1)$ approximation factor and polynomial running time for the bi-criteria $k$-CSS$_{p,2}$ presented in Algorithm 4, given in **Theorem 5**.

**Theorem 5** (Bicriteria $O(1)$-Approximation Algorithm for $k$-CSS$_{p,2}$)**.** *Let $A \in \mathbb{R}^{d \times n}$ and $k \in \mathbb{N}$. There exists an algorithm that runs in $(nnz(A) + d^2) \cdot k\mathrm{poly}(\log k)$ time and outputs a rescaled subset of columns $U \in \mathbb{R}^{d \times \widetilde{O}(k)}$ of $A$ and a right factor $V \in \mathbb{R}^{\widetilde{O}(k) \times n}$ for which $V = \min_V |UV - A|_{p,2}$, such that with probability $1 - o(1)$,*

$$
|UV - A|_{p,2} \leq O(1) \cdot \min_{\text{rank-k } A_k}|A_k - A|_{p,2}
$$

*Proof.* **Approximation Factor** First notice that the minimizer $\hat{X}$ of $|\hat{X}SAS' - AS'|_{p,2}$ has to be in the column span of $AS'$. Thus we can write $\hat{X} = (AS')Y$ for some matrix $Y$. By **Theorem 5.8**,

$$
|\hat{X}SA - A|_{p,2} = |(AS')YSA - A|_{p,2} \leq \Theta(1)\min_{\text{rank-k } A_k}|A - A_k|_{p,2}
$$

We denote $YSA = V$. We take the left factor $U = AS'$ and solving for $\min_V |UV - A|_{p,2}$ will give us a $\Theta(1)$ approximation to $\min_{\text{rank-k } A_k}|A - A_k|_{p,2}$. A good minimizer for the right factor $V$ in the Euclidean space is $V = (AS')^\dagger A$. This concludes our result. Notice that since $S'$ is a sampling matrix with $\widetilde{O}(k)$ columns, we get a rank-$k$ left factor $U$ as a subset of columns of $A$ as desired.

**Running time** First notice that $S$ is a sparse embedding matrix with $\mathrm{poly}(\log k)$ non-zero entries. Thus computing $SA$ takes time $nnz(A) \cdot \mathrm{poly}(\log k)$. By Cohen & Peng (2015), computing the Lewis weights of $SA$ takes time $nnz(SA) + \mathrm{poly}(k) \leq nnz(A)\mathrm{poly}(\log k) + \mathrm{poly}(k)$, and computing the output left factor $U = AS'$ takes time $nnz(A)$. Computing $(AS')^\dagger$ takes time $d^2 \cdot k\mathrm{poly}(\log k)$. Computing the right factor $V = (AS')^\dagger A$ takes $nnz(A)k\mathrm{poly}(\log k)$. Therefore, the overall running time is $(nnz(A) + d^2) \cdot k\mathrm{poly}(\log k)$.

$\square$

## D ANALYSIS FOR GREEDY $k$-CSS$_{1,2}$

We propose a greedy algorithm for selecting columns in $k$-CSS$_{1,2}$ presented in Algorithm **??**. We give a detailed analysis on the first additive approximation compared to the best possible subset of columns for Greedy $k$-CSS$_{1,2}$. Our analysis is inspired by the analysis of Greedy $k$-CSS$_2$ for the Frobenius norm in Altschuler et al. (2016). We then describe how the running time of Algorithm **??** can be improved to $\widetilde{O}(\frac{n}{\sigma\epsilon} \cdot F)$, where $F$ is the running time required to evaluate $\min_V |A_{T \cup j}V - A|_{1,2}$ for a fixed $j \in \overline{T}$ (note that this running time is $O(nk^2 + knd)$ if we evaluate this by computing the pseudo-inverse of $A_{T \cup j}$). This improvement in the running time is obtained by randomly sampling candidate columns from $\overline{T}$ and adding the best of these randomly sampled columns to $A_T$, rather than trying all columns of $T$ — this method was previously used in Altschuler et al. (2016) for Frobenius norm column subset selection, where it is referred to as "Lazier-than-lazy Greedy," and the general approach was first introduced in Mirzasoleiman et al. (2015). This version of the greedy algorithm is shown in Algorithm 2, and we show both versions of the greedy algorithm below for convenience.

---

**Algorithm 5** Greedy $k$-CSS$_{1,2}$. Here we denote the set of selected columns $T$ from $A$ by $A_T$ and the set of unselected columns by $A_{\overline{T}}$.

---

**Input:** The data matrix $A \in \mathbb{R}^{d \times n}$, the number of iterations $r \leq n$.
**Output:** A subset of columns $A_T$ from $A$, where $|T| = r$.
$A_T \leftarrow \emptyset$
**for** $i = 1$ **to** $r$ **do**
    Column $j^* \leftarrow \arg\min_{j \in A_{\overline{T}}} (\min_V |A_{T \cup j} V - A|_{1,2})$
    $A_T \leftarrow A_{T \cup j^*}$
**end for**

---

---

**Algorithm 6** Lazier-than-lazy Greedy $k$-CSS$_{1,2}$. This version of the greedy algorithm is based on Section 5.2 of Altschuler et al. (2016).

---

**Input:** The data matrix $A \in \mathbb{R}^{d \times n}$, the number of iterations $r \leq n$, a parameter $\delta \in (0, 1)$.
**Output:** A subset of columns $A_T$ from $A$, where $|T| = r$.
$A_T \leftarrow \emptyset$
**for** $i = 1$ **to** $r$ **do**
    $\overline{T} \leftarrow$ A subset of $\frac{n \log(1/\delta)}{k}$ columns of $A$, each selected uniformly at random (excluding the columns whose indices are in $T$)
    Column $j^* \leftarrow \arg\min_{j \in A_{\overline{T}}} (\min_V |A_{T \cup j} V - A|_{1,2})$
    $A_T \leftarrow A_{T \cup j^*}$
**end for**

---

First we analyze Algorithm 4, then show how this analysis can be extended to Algorithm 6. We first show in **Lemma** 7.2 an improvement of the utility function with one additional column when projecting a single vector, based on **Lemma** 7.1 from Altschuler et al. (2016). We then show an improvement of the utility function when projecting a matrix in **Lemma** 7.3, by applying **Lemma** 7.2 and Jensen's Inequality, following the analysis in Altschuler et al. (2016). Finally, we conclude our analysis for Greedy $k$-CSS$_{1,2}$ in **Theorem** 7.

**Notation**   Consider the input matrix $A \in \mathbb{R}^{d \times n}$ ($n \gg d$). Let $B$ be the matrix of normalized columns of $A$, where the $j$-th column of $B$ is $B_{*j} = A_{*j} / |A_{*j}|_2$. Let $\pi_T : \mathbb{R}^d \to \mathbb{R}^d$ be the projection onto the column span of $A_T$ or equivalently $B_T$. Let $\sigma_{\min}(M)$ denote the minimum singular value of some matrix $M$.

To aid our analysis, we define a utility function as follows, inspired by Altschuler et al. (2016). For a subset $T \subset [n]$ and a matrix $M \in \mathbb{R}^{d \times t}$ (or a vector $M \in \mathbb{R}^d$),

$$\Phi_M(T) = |M|_{1,2} - |M - \pi_T M|_{1,2} = \sum_{i=1}^{t} \left( |M_{*i}|_2 - |M_{*i} - \pi_T M_{*i}|_{1,2} \right) = \sum_{i=1}^{t} \Phi_{M_{*i}}(T)$$

Observe that as the number of columns selected and added to $T$ increases, we get a more accurate estimation of $M$ and thus the approximation error $|M - \pi_T M|_{1,2}$ decreases, which results in an increase in the utility function $\Phi_M(T)$.

**Lemma 7.1.** *Let* $S, T \subset [n]$ *be two sets of column indices, with* $|\pi_S u|_2 \geq |\pi_T u|_2$ *for some vector* $u \in \mathbb{R}^d$. *Then,*

$$\sum_{i=1}^{k} \left( |\pi_{T_i'} u|_2^2 - |\pi_T u|_2^2 \right) \geq \sigma_{min}(B_S)^2 \frac{(|\pi_S u|_2^2 - |\pi_T u|_2^2)^2}{4 |\pi_S u|_2^2}$$

*Proof.* Lemma 2 from Altschuler et al. (2016), except that we replace the condition for $S$ and $T$, i.e., $\Phi_u(S) \geq \Phi_u(T)$ in Altschuler et al. (2016) with $|\pi_S u|_2 \geq |\pi_T u|_2$. The two conditions are equivalent, since

$$\Phi_u(S) \geq \Phi_u(T)$$
$$\Leftrightarrow |u - \pi_S u|_2 \leq |u - \pi_T u|_2$$
$$\Leftrightarrow |u|_2^2 - |\pi_S u|_2^2 \leq |u|_2^2 - |\pi_T u|_2^2$$
$$\Leftrightarrow |\pi_S u|_2 \geq |\pi_T u|_2$$

$\square$

**Lemma 7.2.** *(Utility Improvement by Projecting a Single Vector) Let $S, T \subset [n]$ be two sets of column indices, with $\Phi_u(S) \geq \Phi_u(T)$ for some vector $u \in \mathbb{R}^d$. Let $k = |S|$. For $i \in [k]$, let $T_i' = T \cup \{i\}$. Then,*

$$\sum_{i=1}^{k} \left( \Phi_u(T_i') - \Phi_u(T) \right) \geq \sigma_{min}(B_S)^2 \frac{(\Phi_u(S) - \Phi_u(T))^3}{16 \Phi_u(S)^2}$$

*Proof.* We define a function for convenience in the analysis $g : (-\infty, |u|_2^2] \to \mathbb{R}_{\geq 0}$ by $g(x) = \sqrt{|u|_2^2 - x}$. Note that $|g'(x)| = \frac{1}{2\sqrt{|u|_2^2 - x}}$.

$$\sum_{i=1}^{k} \left( \Phi_u(T_i') - \Phi_u(T) \right) = \sum_{i=1}^{k} \left( |u - \pi_T u|_2 - |u - \pi_{T_i'} u|_2 \right) \qquad \text{By definition of } \Phi$$

$$= \sum_{i=1}^{k} \left( \sqrt{|u|_2^2 - |\pi_T u|_2^2} - \sqrt{|u|_2^2 - |\pi_{T_i'} u|_2^2} \right) \qquad \text{By Pythagorean Theorem}$$

$$= \sum_{i=1}^{k} \left( g(|\pi_T u|_2^2) - g(|\pi_{T_i'} u|_2^2) \right)$$

$$\geq \sum_{i=1}^{k} |g'(|\pi_T u|_2^2)| \left( |\pi_{T_i'} u|_2^2 - |\pi_T u|_2^2 \right) \qquad \text{By the Mean Value Theorem}$$

$$= \sum_{i=1}^{k} \frac{1}{2\sqrt{|u|_2^2 - |\pi_T u|_2^2}} \left( |\pi_{T_i'} u|_2^2 - |\pi_T u|_2^2 \right)$$

$$= \frac{1}{2|u - \pi_T u|_2} \sum_{i=1}^{k} \left( |\pi_{T_i'} u|_2^2 - |\pi_T u|_2^2 \right) \qquad \text{By Pythagorean Theorem}$$

$$\geq \frac{1}{2|u - \pi_T u|_2} \cdot \sigma_{min}(B_S)^2 \frac{(|\pi_S u|_2^2 - |\pi_T u|_2^2)^2}{4|\pi_S u|_2^2} \qquad \text{By \textbf{Lemma} 7.1}$$

$$= \sigma_{min}(B_S)^2 \frac{(|u - \pi_T u|_2^2 - |u - \pi_S u|_2^2)^2}{8 \cdot |\pi_S u|_2^2 |u - \pi_T u|_2} \qquad \text{By Pythagorean Theorem}$$

$$= \frac{\sigma_{min}(B_S)^2}{8 \cdot |\pi_S u|_2^2 |u - \pi_T u|_2}$$
$$\cdot (|u - \pi_T u|_2 - |u - \pi_S u|_2)^2$$
$$\cdot (|u - \pi_T u|_2 + |u - \pi_S u|_2)^2$$

$$\geq \sigma_{min}(B_S)^2 \frac{(\Phi_u(S) - \Phi_u(T))^2 \cdot |u - \pi_T u|_2}{8 \cdot |\pi_S u|_2^2} \qquad \text{Since } |u - \pi_S u|_2 \geq 0$$

We now lower bound $\frac{|u - \pi_T u|_2}{|\pi_S u|_2^2}$ as follows.

$$\frac{|u - \pi_T u|_2}{|\pi_S u|_2^2} = \frac{|u - \pi_T u|_2}{|u|_2^2 - |u - \pi_S u|_2^2} \qquad \text{By Pythagorean Theorem}$$

$$= \frac{|u - \pi_T u|_2}{|u|_2 - |u - \pi_S u|_2} \cdot \frac{1}{|u|_2 + |u - \pi_S u|_2}$$

$$= \frac{1}{\Phi_u(S)} \cdot \frac{|u - \pi_T u|_2}{|u|_2 + |u - \pi_S u|_2} \qquad \text{By definition of } \Phi$$

$$\geq \frac{1}{2\Phi_u(S)} \cdot \frac{|u - \pi_T u|_2}{|u|_2} \qquad \text{Since } |u - \pi_S u|_2 \leq |u|_2$$

$$= \frac{1}{2\Phi_u(S)} \cdot \frac{|u|_2 - \Phi_u(T)}{|u|_2} \qquad \text{By definition of } \Phi$$

$$= \frac{1}{2\Phi_u(S)} \cdot \left( 1 - \frac{\Phi_u(T)}{|u|_2} \right)$$

$$\geq \frac{1}{2\Phi_u(S)} \cdot \left( 1 - \frac{\Phi_u(T)}{\Phi_u(S)} \right) \qquad \text{Since } \Phi_u(S) \leq |u|_2$$

$$= \frac{(\Phi_u(S) - \Phi_u(T))}{2\Phi_u(S)^2}$$

Therefore,

$$\sum_{i=1}^{k} \left( \Phi_u(T_i') - \Phi_u(T) \right) \geq \sigma_{min}(B_S)^2 \frac{(\Phi_u(S) - \Phi_u(T))^2 \cdot |u - \pi_T u|_2}{8 \cdot |\pi_S u|_2^2}$$

$$\geq \sigma_{min}(B_S)^2 \frac{(\Phi_u(S) - \Phi_u(T))^3}{16\Phi_u(S)^2}$$

$\square$

**Lemma 7.3.** *(Utility Improvement by Projecting a Matrix) Let $A \in \mathbb{R}^{d \times n}$, and $T, S \subset [n]$ be two sets of column indices, with $\Phi_A(S) \geq \Phi_A(T)$. Furthermore, let $k = |S|$. Then, there exists a column index $i \in S$ such that*

$$\Phi_A(T \cup \{i\}) - \Phi_A(T) \geq \sigma_{min}(B_S)^2 \frac{(\Phi_A(S) - \Phi_A(T))^3}{16k\Phi_A(S)^2}$$

*Proof.* The proof mostly follows the proof of Lemma 1 in Altschuler et al. (2016). We combine Lemma 7.2 with Jensen's inequality to conclude an improvement of the utility function with one additional column when projecting a matrix instead of a single column.

For $j \in [n]$, we define $\delta_j = \min(1, \frac{\Phi_{A_{*j}}(T)}{\Phi_{A_{*j}}(S)})$. Note that $\delta_j$ is 1 if the $j$-th column $A_{*j}$ has a larger projection onto $B_T$ than $B_S$, and $\frac{\Phi_{A_{*j}}(T)}{\Phi_{A_{*j}}(S)}$ otherwise. Let $k = |S|$. For $i \in [k]$, let $T_i' = T \cup \{i\}$.

$$\frac{1}{\sigma_{min}(B_S)^2} \sum_{i=1}^{k} \left( \Phi_A(T_i') - \Phi_A(T) \right) = \frac{1}{\sigma_{min}(B_S)^2} \sum_{j=1}^{n} \sum_{i=1}^{k} \left( \Phi_{A_{*j}}(T_i') - \Phi_{A_{*j}}(T) \right) \quad \text{By definition of } \Phi$$

$$\geq \sum_{j=1}^{n} \frac{(1 - \delta_j)^3}{16} \cdot \Phi_{A_{*j}}(S) \quad \text{By \textbf{Lemma} 7.2}$$

$$= \frac{\Phi_A(S)}{16} \sum_{j=1}^{n} (1 - \delta_j)^3 \cdot \frac{\Phi_{A_{*j}}(S)}{\sum_{i=1}^{n} \Phi_{A_{*i}}(S)} \quad \text{Note } \Phi_A(S) = \sum_{i=1}^{n} \Phi_{A_{*i}}(S)$$

$$\geq \frac{\Phi_A(S)}{16} \left( \sum_{j=1}^{n} (1 - \delta_j) \cdot \frac{\Phi_{A_{*j}}(S)}{\sum_{i=1}^{n} \Phi_{A_{*i}}(S)} \right)^3 \quad \text{By Jensen's Inequality}$$

$$= \frac{1}{16\Phi_A(S)^2} \left( \sum_{j=1}^{n} (1 - \delta_j) \cdot \Phi_{A_{*j}}(S) \right)^3 \quad \text{Since } 1 - \delta_j \geq 1 - \frac{\Phi_{A_{*j}}(T)}{\Phi_{A_{*j}}(S)}$$

$$\Rightarrow (1 - \delta_j) \cdot \Phi_{A_{*j}}(S)$$
$$\geq \Phi_{A_{*j}}(S) - \Phi_{A_{*j}}(T)$$

$$\geq \frac{1}{16\Phi_A(S)^2} \left( \sum_{j=1}^{n} (\Phi_{A_{*j}}(S) - \Phi_{A_{*j}}(T)) \right)^3$$

$$= \frac{(\Phi_A(S) - \Phi_A(T))^3}{16\Phi_A(S)^2}$$

Hence,

$$\sum_{i=1}^{k} \left( \Phi_A(T_i') - \Phi_A(T) \right) \geq \sigma_{min}(B_S)^2 \frac{(\Phi_A(S) - \Phi_A(T))^3}{16\Phi_A(S)^2}$$

This implies there is at least one column of $B_S$, with index $i \in S$, such that when $i$ is added to $T$, the utility function $\Phi_A(T)$ increases by at least $\frac{1}{k} \cdot \sigma_{min}(B_S)^2 \frac{(\Phi_A(S) - \Phi_A(T))^3}{16\Phi_A(S)^2}$. $\square$

**Theorem 7.** *Let $A \in \mathbb{R}^{d \times n}$ be the data matrix and $k \in \mathbb{N}$ be the desired rank. Let $A_S$ be the best possible subset of $k$ columns, i.e., $A_S = \arg\min_{A_S} \min_V |A_S V - A|_{1,2}$. Let $\sigma$ be the minimum non-zero singular*

*value of the matrix $B$ of normalized columns of $A_S$, (the $j$-th column of $B$ is $B_{*j} = (A_S)_{*j}/|(A_S)_{*j}|_2$). Then, if $T \subset [n]$ is the subset of columns selected by Greedy $k$-$CSS_{1,2}$, the following holds with $|T| = \Omega(\frac{k}{\sigma^2 \epsilon^2})$,*

$$\min_V |A_T V - A|_{1,2} \leq (1 - \epsilon) \min_{S \subset [n], |S|=k, V \in \mathbb{R}^{k \times n}} |A_S V - A|_{1,2} + \epsilon |A|_{1,2}$$

*Proof.* The proof follows the one of Theorem 1 in Altschuler et al. (2016).

Let $T_t$ be the subset of columns of $B$ selected by Greedy $k$-$CSS_{1,2}$ after $t$ iterations. Notice that $T_0 = \varnothing$. In addition, define $F = \Phi_A(S) = \Phi_A(S) - \Phi_A(T_0)$, $\Delta_0 = F$, and $\Delta_i = \frac{\Delta_0}{2^i}$ for $i \in \mathbb{N}$.

Let $\Delta_i \geq \Phi_A(S) - \Phi_A(T_t) \geq \Delta_{i+1} = \frac{\Delta_i}{2}$. Our goal is to bound the number of iterations needed for the gap between $\Phi_A(S)$ and $\Phi_A(T_t)$ to become less than $\frac{\Delta_i}{2}$.

Consider some iteration $s$ for which $\Phi_A(S) - \Phi_A(T_s) \geq \frac{\Delta_i}{2}$. The improvement of the utility function after adding a column of $B$ to $T_s$ through greedy selection is at least the improvement of the utility function after adding the best column of $B_S$ to $T_s$. By **Lemma** 7.3, this is at least

$$\sigma^2 \frac{(\Phi_A(S) - \Phi_A(T))^3}{16 k \Phi_A(S)^2} = \sigma^2 \cdot \frac{\Delta_i^3}{16 \cdot 8 \cdot k \cdot F^2} = \frac{\sigma^2 \Delta_i^3}{128 k F^2}.$$

If the gap $\Phi_A(S) - \Phi_A(T_t)$ after $t$ iterations is at most $\Delta_i$ and at least $\frac{\Delta_i}{2}$, then after at most $\frac{64 k F^2}{\sigma^2 \Delta_i^2}$ iterations, the gap becomes at most $\frac{\Delta_i}{2}$.

We can use this to bound the number of iterations required for the gap to become at most $\epsilon F$. Take $N \in \mathbb{N}$ such that $\Delta_{N+1} \leq \epsilon F \leq \Delta_N$. Then the number of iterations required for the gap to become at most $\Delta_{N+1}$ is at most

$$\sum_{i=0}^{N} \frac{64 k F^2}{\sigma^2 \Delta_i^2} = \frac{64 k F^2}{\sigma^2} \sum_{i=0}^{N} \frac{1}{\Delta_i^2}$$

$$= \frac{64 k F^2}{\sigma^2 \Delta_{N+1}^2} \sum_{i=0}^{N} \frac{1}{4^{N+1-i}} \qquad \qquad \text{Since } \Delta_{N+1} = \frac{\Delta_i}{2^{N+1-i}}$$

$$\leq \frac{256 k}{3 \sigma^2 \epsilon^2} \qquad \qquad \text{Since } \Delta_{N+1} \geq \frac{\epsilon F}{2} \text{ and } \sum_{i=0}^{N} \frac{1}{4^{N+1-i}} \leq \frac{1}{3}$$

Therefore, after $|T| = \Omega(\frac{k}{\delta^2 \epsilon^2})$ iterations, we have

$$\Phi_A(S) - \Phi_A(T) \leq \epsilon \Phi_A(S)$$
$$\Rightarrow |A|_{1,2} - |A - \pi_S A|_{1,2} - (|A|_{1,2} - |A - \pi_T A|_{1,2}) \leq \epsilon(|A|_{1,2} - |A - \pi_S A|_{1,2})$$
$$\Rightarrow |A - \pi_T A|_{1,2} \leq (1 - \epsilon)|A - \pi_S A|_{1,2} + \epsilon |A|_{1,2}$$

Since $S$ is the set of indices for the best possible subset of $k$ columns, the above is equivalent to

$$\min_V |A_T V - A|_{1,2} \leq (1 - \epsilon) \min_{S \subset [n], |S|=k, V \in \mathbb{R}^{k \times n}} |A_S V - A|_{1,2} + \epsilon |A|_{1,2}$$

$\square$

We now analyze Algorithm 2, based on the analysis of the Lazier-than-lazy greedy heuristic in Altschuler et al. (2016). The first step is the following lemma based on Lemma 6 of Altschuler et al. (2016), which shows that in expectation, the utility improves by a large amount on each iteration.

**Lemma 7.4** (Expected Increase in Utility — Based on Lemma 6 of Altschuler et al. (2016)). *Let $A \in \mathbb{R}^{d \times n}$, and let $T, S \subset [n]$ be two sets of column indices, with $k := |S|$ and $\Phi_A(S) \geq \Phi_A(T)$. Let $\overline{T}$ be a set of $\frac{n \log(1/\delta)}{k}$ column indices of $A$, chosen uniformly at random from $[n] \setminus T$. Then,*

$$E[\max_{i \in \overline{T}} \Phi_A(T \cup \{i\})] - \Phi_A(T) \geq (1 - \delta) \cdot \sigma_{min}(B_S)^2 \cdot \frac{(\Phi_A(S) - \Phi_A(T))^3}{16 k \Phi_A(S)^2}$$

*Proof.* The proof is nearly identical to the proof of Lemma 6 of Altschuler et al. (2016) — we include the full proof for completeness. The first step in the proof is showing that $\overline{T} \cap (S \setminus T)$ is nonempty with high probability. Then, by conditioning on $\overline{T} \cap (S \setminus T)$ being nonempty, we can show that the expected increase in utility is large. For the purpose of this analysis, we assume that the columns of $\overline{T}$ are sampled independently with replacement. At the end of the proof, we discuss sampling the columns of $\overline{T}$ without replacement.

First, observe that

$$\Pr[\overline{T} \cap (S \setminus T) = \varnothing] = \prod_{t=1}^{O\left(\frac{n \log(1/\delta)}{k}\right)} \left(1 - \frac{|S \setminus T|}{n - |T|}\right)$$

$$= \left(1 - \frac{|S \setminus T|}{n - |T|}\right)^{O\left(\frac{n \log(1/\delta)}{k}\right)}$$

$$\leq e^{-\frac{|S \setminus T|}{n - |T|} \cdot \frac{n \log(1/\delta)}{k}} \qquad \text{By } 1 - x \leq e^{-x}$$

$$\leq e^{-\frac{|S \setminus T| \log(1/\delta)}{k}} \qquad \text{Because } n - |T| < n$$

meaning that

$$\Pr[\overline{T} \cap (S \setminus T)] \geq 1 - e^{-\frac{|S \setminus T| \log(1/\delta)}{k}}$$

$$= 1 - \delta^{\frac{|S \setminus T|}{k}}$$

$$\geq (1 - \delta)\frac{|S \setminus T|}{k} \qquad \text{Since } |S \setminus T| \leq k, \text{ and } 1 - \delta^x \geq (1 - \delta)x \text{ for } x, \delta \in [0, 1]$$

Therefore,

$$E[\max_{i \in \overline{T}} \Phi_A(T \cup \{i\}) - \Phi_A(T)]$$

$$\geq \Pr[\overline{T} \cap (S \setminus T) \neq \varnothing] \cdot E\left[\max_{i \in \overline{T}} \Phi_A(T \cup \{i\}) - \Phi_A(T) \middle| \overline{T} \cap (S \setminus T) \neq \varnothing\right]$$

$$\geq (1 - \delta)\frac{|S \setminus T|}{k} \cdot E\left[\max_{i \in \overline{T}} \Phi_A(T \cup \{i\}) - \Phi_A(T) \middle| \overline{T} \cap (S \setminus T) \neq \varnothing\right]$$

$$\geq (1 - \delta)\frac{|S \setminus T|}{k} \cdot E\left[\max_{i \in \overline{T}} \Phi_A(T \cup \{i\}) - \Phi_A(T) \middle| |\overline{T} \cap (S \setminus T)| = 1\right]$$

(Since it is always better for $\overline{T} \cap (S \setminus T)$ to be larger)

$$= (1 - \delta)\frac{|S \setminus T|}{k} \cdot \frac{\sum_{i \in S \setminus T}(\Phi_A(T \cup \{i\}) - \Phi_A(T))}{|S \setminus T|}$$

$$= (1 - \delta) \cdot \frac{\sum_{i \in S}(\Phi_A(T \cup \{i\}) - \Phi_A(T))}{|S|}$$

(Since $\Phi_A(T \cup \{i\}) = \Phi_A(T)$ for $i \in T$)

$$\geq (1 - \delta) \cdot \frac{1}{|S|} \cdot \sigma_{min}(B_S)^2 \frac{(\Phi_A(S) - \Phi_A(T))^2}{16\Phi_A(S)^2}$$

(See the proof of Lemma 7.3.)

This proves the lemma in the case where the columns are sampled with replacement. Now, we discuss what happens when sampling without replacement. Note that the expected increase in utility can only be higher if the columns of $\overline{T}$ are sampled without replacement. Intuitively, this is because if $\overline{T}$ has some repeated columns, then it is always better to replace those repeated columns with other columns of $A$. Thus, for each instance of $\overline{T}$ where some columns are sampled multiple times, we can "move" all of the probability mass from this instance of $\overline{T}$ to other sets $\overline{T}' \subset [n] \setminus T$, which contain $\overline{T}$ but do not have repeated elements. This leads to the uniform distribution on subsets of $[n] \setminus T$ with no repeated elements, i.e., the distribution that results from sampling without replacement. □

Using this lemma, we analyze the convergence of Algorithm 2:

**Theorem 8.** *Let $A \in \mathbb{R}^{d \times n}$ be the data matrix and $k \in \mathbb{N}$ the desired rank. Let $A_S$ be the best subset of $k$ columns, i.e., $A_S = \arg\min_{A_S} \min_V |A_S V - A|_{1,2}$. Let $\sigma$ be the minimum non-zero singular value of the matrix $B$ of normalized columns of $A_S$ (meaning the $j$-th column of $B$ is $B_{*j} = (A_S)_{*j}/|(A_S)_{*j}|_2$). Then, if $T \subset [n]$ is the subset of columns selected by Algorithm 2, the following holds if $|T| = \Omega(\frac{k}{\sigma^2 \epsilon^2})$:*

$$E[\min_V |A_T V - A|_{1,2}] \leq (1 - \epsilon) \min_{S \subset [n], |S| = k, V \in \mathbb{R}^{k \times n}} |A_S V - A|_{1,2} + \epsilon|A|_{1,2}$$

*Proof.* The proof is uses the same strategy as that of Theorem 5 of Altschuler et al. (2016) (and Theorem 7 above), with minor modifications. Let $T_t$ be the subset of columns of $B$ selected by Algorithm 2 after $t$ iterations (in particular, $T_0 = \varnothing$). In addition, let $F = \Phi_A(S) = \Phi_A(S) - \Phi_A(T_0)$, $\Delta_0 = F$, and $\Delta_{i+1} = \frac{\Delta_i}{2}$. Now, fix a time $t$ such that for some $i$, $\Delta_i \geq \Phi_A(S) - \Phi_A(T_t) \geq \Delta_{i+1} = \frac{\Delta_i}{2}$. Then, we bound the number of additional iterations $t'$ needed so that

$$E[\Phi_A(S) - \Phi_A(T_{t+t'}) \mid T_t] < \Delta_{i+1}$$

For convenience, for each $k \geq 0$, define $E_k := \Phi_A(T_{t+k})$. Then, our goal is to find $t'$ such that
$$\Phi_A(S) - E_{t'} < \Delta_{i+1}$$
However, observe that from Lemma 7.4 above, we obtain

$$
\begin{aligned}
E_{k+1} - E_k &= E\Big[\Phi_A(T_{t+k+1}) - \Phi_A(T_{t+k})\Big|T_t\Big] \\
&= E\Big[E\big[\Phi_A(T_{t+k+1}) - \Phi_A(T_{t+k})\big|T_{t+k}\big]\Big|T_t\Big] && \text{By } E[E[X|Y]] = E[X] \\
&\geq E\Big[(1-\delta)\cdot\sigma_{min}(B_S)^2 \cdot \frac{(\Phi_A(S) - \Phi_A(T_{t+k}))^3}{16k\Phi_A(S)^2}\Big|T_t\Big] && \text{By Lemma 7.4} \\
&= \frac{(1-\delta)\cdot\sigma_{min}(B_S)^2}{16k\Phi_A(S)^2} \cdot E\big[(\Phi_A(S) - \Phi_A(T_{t+k}))^3|T_t\big] \\
&\geq \frac{(1-\delta)\cdot\sigma_{min}(B_S)^2}{16k\Phi_A(S)^2} \cdot \Big(E[\Phi_A(S) - \Phi_A(T_{t+k})|T_t]\Big)^3 && \text{By Jensen's Inequality} \\
&= (1-\delta)\cdot\sigma_{min}(B_S)^2 \cdot \frac{(E[\Phi_A(S)] - E_k)^3}{16k\Phi_A(S)^2}
\end{aligned}
$$

Now, suppose that $\Delta_i \geq \Phi_A(S) - E_s \geq \Delta_{i+1}$, for $s = 0, \ldots, t'-1$. Then, for all such $s$, $E_{s+1} - E_s \geq \frac{(1-\delta)\sigma_{min}(B_S)^2\Delta_{i+1}^3}{16kF^2}$. Summing these inequalities for $s = 0, \ldots, t'-1$, we find that

$$E_{t'} - E_0 \geq \frac{(1-\delta)\sigma_{min}(B_S)^2}{16kF^2} \cdot \Delta_{i+1}^3 \cdot t'$$

and for the increase from $E_0$ to $E_{t'}$ to be greater than $\Delta_{i+1}$, it suffices to have

$$t' \geq \frac{32kF^2}{\Delta_{i+1}^2 \cdot (1-\delta)\sigma_{min}(B_S)^2}$$

In summary, if $\Phi_A(S) - E[\Phi_A(T_t)] \leq \Delta_i$, then in at most $\frac{32kF^2}{\Delta_{i+1}\cdot(1-\delta)\sigma_{min}(B_S)^2}$ iterations, $\Phi_A(S) - E[\Phi_A(T_t)] \leq \Delta_{i+1}$. Thus, if we let $N \in \mathbb{N}$ such that $\Delta_{N+1} \leq \frac{\epsilon}{\sqrt{1-\delta}}F \leq \Delta_N$, then the number of iterations $t$ needed to have $\Phi_A(S) - E[\Phi_A(T_t)] < \Delta_{N+1}$ is at most

$$
\begin{aligned}
\sum_{i=0}^N \frac{32kF^2}{\Delta_{i+1}^2 \cdot (1-\delta)\sigma_{min}(B_S)^2} &= \frac{32kF^2}{(1-\delta)\sigma_{min}(B_S)^2} \sum_{i=0}^N \frac{1}{\Delta_{i+1}^2} \\
&= \frac{32kF^2}{(1-\delta)\sigma_{min}(B_S)^2} \sum_{i=0}^N \frac{1}{4^{N-i}} \cdot \frac{1}{\Delta_{N+1}^2} \\
&\leq \frac{32kF^2}{(1-\delta)\sigma_{min}(B_S)^2} \cdot \frac{4(1-\delta)}{\epsilon^2 F^2} \sum_{i=0}^N \frac{1}{4^{N-i}} && \text{Since } \Delta_{N+1} \geq \frac{\epsilon F}{2\sqrt{1-\delta}} \\
&= \frac{128k}{\sigma_{min}(B_S)^2\epsilon^2} \sum_{i=0}^N \frac{1}{4^i} \\
&\leq \frac{512k}{3\sigma_{min}(B_S)^2\epsilon^2}
\end{aligned}
$$

Thus, after $t = O(\frac{k}{\sigma_{min}(B_S)^2\epsilon^2})$ iterations,

$$\Phi_A(S) - E[\Phi_A(T_t)] \leq \frac{\epsilon}{\sqrt{1-\delta}}\Phi_A(S)$$

meaning

$$|A|_{1,2} - |A - \pi_S A|_{1,2} - E[|A|_{1,2} - |A - \pi_T A|_{1,2}] \leq \frac{\epsilon}{\sqrt{1-\delta}}|A|_{1,2} - \frac{\epsilon}{\sqrt{1-\delta}}|A - \pi_S A|_{1,2}$$

and rearranging gives

$$E[|A - \pi_T A|_{1,2}] \leq \Big(1 - \frac{\epsilon}{\sqrt{1-\delta}}\Big)|A - \pi_S A|_{1,2} + \frac{\epsilon}{\sqrt{1-\delta}}|A|_{1,2}$$

This completes the proof (note that we can select $\delta = \epsilon$, meaning $\frac{1}{\sqrt{1-\delta}} = O(1)$ for $\epsilon < \frac{1}{2}$). $\qquad\square$

# E EXTENSION TO THE STREAMING MODEL

In this section, we describe how our protocol in Algorithm 1 can be made into a 1-pass streaming algorithm for column subset selection in the $\ell_p$ norm. The algorithm is shown in Algorithm 7, and is analyzed in Theorem 9 below. The algorithm and its analysis follow the standard merge-and-reduce framework (see McGregor (2014)).

**Theorem 9** (Analysis of Algorithm 7). *Let $A \in \mathbb{R}^{d \times n}$ and $k \in \mathbb{N}$, and assume Algorithm 7 sees the columns of $A$ one at a time in the stream $\mathcal{S}$. Then, Algorithm 7 returns $U \in \mathbb{R}^{d \times \widetilde{O}(k)}$ such that*

$$\min_{V \in \mathbb{R}^{\widetilde{O}(k) \times n}} \|UV - A\|_p \le \widetilde{O}(k^{1/p-1/2}) \min_{A_k \ rank \ k} \|A_k - A\|_p$$

*with probability 0.9. Moreover, the space complexity of Algorithm 7 is $\widetilde{O}(dk)$.*

*Proof.* Let $r = \widetilde{O}(k)$ be the bi-criteria rank of Algorithm 4. Then, we will be repeatedly applying Lemma 4 with $k$ being equal to $r$, i.e., we will create coresets which preserve the errors when projecting onto all subspaces of dimension $r$.

Now, at every iteration of Algorithm 7, each element $(L, t)$ of $\mathcal{L}$ can be thought of as holding a coreset of the columns of $A$ in some interval $I$ in $[n]$. We prove the following intermediate lemma by induction on $t$:

**Lemma 10.** *Let $B$ be the concatenation of all the sketched columns in $L$ (which have been reweighted by multiple applications of Lemma 4). Then, for all subspaces $V \subset \mathbb{R}^{\widetilde{O}(k)}$ of dimension at most $r$,*

$$\|B - P_V B\|_{p,2} = \left(1 \pm \frac{1}{\log n}\right)^t \|SA_I - P_V SA_I\|_{p,2}$$

*Proof.* We proceed by induction on $t$. The lemma is clear when $t = 0$, since in that case, the columns of $B$ are simply sketched columns of $A$ which have not been re-weighted.

Now, suppose $t > 0$, and suppose the lemma holds for smaller values of $t$. Then, the sketched columns in $L$ must have been obtained as follows: there previously existed two elements $(L_1, t-1)$ and $(L_2, t-1)$ of $\mathcal{L}$, such that if $B_1$ is the concatenation of the sketched columns in $L_1$ and $B_2$ is the concatenation of the sketched columns in $L_2$, and $T$ is a coreset for the concatenation $B_3$ of $B_1$ and $B_2$, then $B = B_3 T$. The sketched columns in $L_1$ and $L_2$ form coresets for two intervals in $[n]$, which we denote by $I_1$ and $I_2$ respectively. Applying Lemma 4, we find that

$$\begin{aligned}
\|B - P_V B\|_{p,2}^p &= \left(1 \pm \frac{1}{\log n}\right)^p \|B_3 - P_V B_3\|_{p,2}^p \\
&= \left(1 \pm \frac{1}{\log n}\right)^p \left(\|B_1 - P_V B_1\|_{p,2}^p + \|B_2 - P_V B_2\|_{p,2}^p\right) \\
&= \left(1 \pm \frac{1}{\log n}\right)^p \cdot \left(1 \pm \frac{1}{\log n}\right)^{p(t-1)} \left(\|SA_{I_1} - P_V SA_{I_1}\|_{p,2}^p + \|SA_{I_2} - P_V SA_{I_2}\|_{p,2}^p\right) \\
&= \left(1 \pm \frac{1}{\log n}\right)^{pt} \cdot \|SA_I - P_V SA_I\|_{p,2}^p
\end{aligned}$$

(1)

where the second and last equalities are because the $p^{th}$ power of the $\ell_{p,2}$ norm of a matrix decomposes across the columns of the matrix, and the third equality is by the induction hypothesis. By taking $p^{th}$ roots, we find that

$$\|B - P_V B\|_{p,2}^p = \left(1 \pm \frac{1}{\log n}\right)^t \|SA_I - P_V SA_I\|_{p,2}^p$$

This proves the lemma. $\square$

Now, if $L$ is the unique element of $\mathcal{L}$ remaining at the end of the for loop in Algorithm 7, let $B$ be the concatenation of the sketched columns in $L$. Then, $L$ is a coreset for all the columns of $A$, and by the above lemma, the distortion of $L$ is at most $(1 \pm \frac{1}{\log n})^{\log n} \in [\frac{1}{e}, e]$ (since $t \le \log n$ — note that, as in all applications of the merge-and-reduce framework, the coresets contained in $\mathcal{L}$ over the course of the algorithm form a binary tree, with the leaf nodes being contiguous intervals of length $\widetilde{O}(k)$). Hence, for all subspaces $V$ of dimension at most $r$,

$$\|B - P_V B\|_{p,2} = \Theta(1) \|SA - P_V SA\|_{p,2}$$

and in particular, if $M \in \mathbb{R}^{\widetilde{O}(k) \times \widetilde{O}(k)}$ is the matrix formed by concatenating the columns of $B$ selected by running Algorithm 4, then

$$\|SA - P_M SA\|_{p,2} \le \Theta(1) \min_{T \subset [n], |T| \le k} \|SA - P_T SA\|_{p,2}$$

where $P_M$ on the left-hand side denotes the projection onto the column span of $M$, and $P_T$ on the right-hand side denotes the projection to the column span of $SA_T$. Hence, by Lemma 1,

$$\|SA - P_M SA\|_p \leq \widetilde{O}(k^{1/p-1/2}) \min_{T \subset [n], |T| \leq k} \|SA - P_T SA\|_p$$

meaning that if $U \in \mathbb{R}^{d \times \widetilde{O}(k)}$ is the matrix whose columns are the unsketched columns corresponding to $M$, then

$$\min_{V \in \mathbb{R}^{\widetilde{O}(k) \times d}} \|SA - SUV\|_p \leq \min_{T \subset [n], |T| \leq k, V \in \mathbb{R}^{k \times d}} \|SA - SA_T V\|_p$$

and by Lemmas 2 and 3, this means

$$\min_{V \in \mathbb{R}^{\widetilde{O}(k) \times d}} \|A - UV\|_p \leq \min_{T \subset [n], |T| \leq k, V \in \mathbb{R}^{k \times d}} \|A - A_T V\|_p$$

Now, we analyze the space complexity of Algorithm 7. Note that at any iteration of the algorithm, $\mathcal{L}$ can hold at most $\log n$ lists of columns (since each element of $\mathcal{L}$ is a coreset corresponding to an interval of column indices in $[n]$ of size $2^k$ for some $k \in \mathbb{N}$, and if two adjacent coresets are of the same size then they will have been merged, so all coresets in $\mathcal{L}$ are of different sizes). Each list of columns is of size at most $\widetilde{O}(k)$, and each column has $d$ entries, meaning the amount of space used in any iteration is at most $\widetilde{O}(dk)$. $\qquad\square$

# F   COMPARISON OF OUR PROTOCOL WITH THE DISTRIBUTED GREEDY PROTOCOL OF ALTSCHULER ET AL. (2016) FOR THE FROBENIUS NORM

In this section, we perform an empirical comparison of our protocol with the distributed greedy protocol for column subset selection in the Frobenius norm due to Altschuler et al. (2016).

## F.1   DISTRIBUTED GREEDY PROTOCOL OF ALTSCHULER ET AL. (2016)

We first recall the distributed protocol of Altschuler et al. (2016), in Algorithm 8. Note that it is a bi-criteria algorithm, i.e., more than $k$ columns are selected, and in Altschuler et al. (2016) it is shown that this gives a good approximation relative to the optimal column subset for the Frobenius norm.

Naïvely, this would require a large communication cost, since the entire data matrix $A$ would have to be communicated between the servers and the coordinator, in order for them to perform the calls $\text{GREEDY}(A, T_i, \frac{32k}{\sigma})$ and $\text{GREEDY}(A, T, \frac{12k}{\sigma})$. Instead, Altschuler et al. (2016) uses the technique of Projection-Cost Preserving sketches, developed in Cohen et al. (2015), in which a key result is as follows:

**Theorem 11** (Projection-Cost Preserving Sketches - Theorem 4 of Cohen et al. (2015), as stated in Altschuler et al. (2016)). *Let $R$ be a random matrix with $n$ rows and $n' = O(\frac{k+\log(\frac{1}{\delta})}{\epsilon^2})$ columns, where each entry is independently and uniformly set to $\pm\sqrt{\frac{1}{n'}}$. Then for any matrix $A \in \mathbb{R}^{d \times n}$, with probability $1 - O(\delta)$, the following holds: for some constant $c \geq 0$, for any $k$-dimensional subspace $U$ of $\mathbb{R}^d$, if $P_U \in \mathbb{R}^{d \times d}$ is the corresponding projection matrix, then*

$$(1 - \epsilon)|P_U A|_2^2 \leq |P_U(AR)|_2^2 + c \leq (1 + \epsilon)|P_U A|_2^2$$

*that is, $AR$ is a Projection-Cost Preserving sketch for $A$. In other words, $R$ preserves the cost of all projections of $A$ onto rank-$k$ subspaces of $\mathbb{R}^d$.*

Hence, when each server performs the call $\text{GREEDY}(A, T_i, \frac{32k}{\sigma})$, in place of $A$, it can use a projection-cost preserving sketch $AR$ for $A$ (and the coordinator can similarly use $AR$ when it makes the call $\text{GREEDY}(A, T, \frac{12k}{\sigma})$). This is because the calls to GREEDY repeatedly compute the cost of the projection of the matrix $A$ onto various column subsets of $T_i$ or $T$, and hence all that is needed is a way to efficiently compute the cost of the projection of $A$ onto various subsets of size $O(\frac{k}{\sigma})$.

$AR$ can be computed without communicating the entire matrix $A$ between the servers, as follows:

- First, the coordinator generates $R \in \mathbb{R}^{n \times n'}$ (where $n' = O(k)$) as specified in Theorem 11. This is sent to all the servers (which can be done with negligible cost if a random seed is sent, for example).

- Suppose the $i^{th}$ server holds $A_i$, which consists of columns with indices $s_i$ through $t_i$ of $A$. Then, if $R_i$ is the $(t_i - s_i + 1) \times n'$ submatrix of $R$ consisting of rows $s_i$ through $t_i$, then the $i^{th}$ server can send $A_i R_i$ to the coordinator. Since $A_i R_i$ is a $d \times O(k)$ matrix, this step takes $O(sdk)$ communication.

- Finally, by definition of matrix multiplication, $AR = \sum_{i=1}^{s} A_i R_i$. The coordinator performs this computation and sends $AR$ to each server. This step also takes $O(sdk)$ communication.

**Algorithm 7** 1-pass streaming algorithm for $k$-CSS$_p$. $\mathcal{L}$ is a collection of coresets of columns. Over the course of the algorithm, each element $L \in \mathcal{L}$ will represent a contiguous subset of $2^t$ columns, for some $t \in [\log n]$ — to determine when two coresets should be merged, we will also keep track of the size of each coreset in $\mathcal{L}$. Hence, each element of $\mathcal{L}$ is of the form $(L, t)$ where $L$ is a list of sketched columns (and their corresponding unsketched columns) and $t$ is the number of times this list has been involved in a merging operation.

---

**Input:** A stream $\mathcal{S}$ in which the columns of the data matrix $A \in \mathbb{R}^{d \times n}$ arrive one at a time, and the target rank $k \in \mathbb{N}$

**Output:** The left factor $U \in \mathbb{R}^{d \times \widetilde{O}(k)}$

$S \leftarrow$ An $\widetilde{O}(k) \times d$ random matrix with i.i.d. standard $p$-stable entries

$\epsilon \leftarrow \frac{1}{\log n}$

$\delta \leftarrow O(\frac{1}{n})$

$f \leftarrow \widetilde{O}(k)$

$\mathcal{L} \leftarrow \varnothing$

**for** Each column $A_{*j}$ that arrives from $\mathcal{S}$ **do**

  **if** $\mathcal{L}$ is empty. **then**

    $L \leftarrow \{\}$, where $\{\}$ is the empty list.

  **else if** The last element $(J, t)$ of $\mathcal{L}$ is such that $t = 0$ (i.e., it has been merged 0 times). **then**

    $L \leftarrow J$

    Remove $(J, t)$ from the end of $\mathcal{L}$.

  **else**

    $L \leftarrow \{\}$

  **end if**

  $L \leftarrow L \cup \{(SA_{*j}, A_{*j})\}$

  $\mathcal{L} \leftarrow \mathcal{L} \cup (L, 0)$

  /* Now, we merge coresets in $\mathcal{L}$ as much as possible. */

  **while** True **do**

    Exit this while loop if $\mathcal{L}$ has only 1 element.

    $(L, t) \leftarrow$ last element of $\mathcal{L}$ — remove this from $\mathcal{L}$.

    $(L', t') \leftarrow$ last element of $\mathcal{L}$ — remove this from $\mathcal{L}$.

    If $t \neq t'$, then add $(L', t')$ back to $\mathcal{L}$, and add $(L, t)$ back to $\mathcal{L}$. Then, exit this while loop.

    $C \leftarrow$ A coreset of $L \cup L'$, computed as specified in Lemma 4 — the $k$ in the statement of Lemma 4 will be the bicriteria rank of Algorithm 4, which is $\widetilde{O}(k)$. The parameters $\delta$ and $\epsilon$ will be as specified at the beginning of this algorithm. (Only compute the coreset of the sketched columns — for those which are included in the coreset, include the corresponding unsketched columns as well (but without re-weighting them)).

    $\mathcal{L} \leftarrow (C, t + 1)$ — note that for each of the re-scaled columns that are included in $C$, we include their original indices in $A$ as well.

  **end while**

**end for**

$L \leftarrow$ The unique element of $\mathcal{L}$

$L' \leftarrow$ The result of running Algorithm 4 on the sketched columns in $L$ — for each of these sketched columns, store the corresponding unsketched column as well.

$U \leftarrow$ The $d \times \widetilde{O}(k)$ matrix whose columns are the unsketched columns in $L'$

**return** $U$

---

**Algorithm 8** Distributed Greedy Column Subset Selection for the Frobenius Norm (Algorithm 2 of Altschuler et al. (2016)). Throughout this protocol, GREEDY$(A, T, r)$ denotes a single-machine procedure, which does the following: if $A \in \mathbb{R}^{d \times n}$ is a data matrix, and $T \in \mathbb{R}^{d \times t}$ is a set of columns (not necessarily of $A$) then a subset $S$ of $r$ columns of $T$ is constructed iteratively over $r$ steps, such that at each step, the new column of $T$ to add to $S$ is greedily chosen — that is, the chosen column increases $|\pi_S A|_2^2$ the most, or equivalently, decreases $|A - \pi_S A|_2^2$ the most. In other words, it is the same as our Algorithm **??**, but for the Frobenius norm rather than the $\ell_{1,2}$-norm.

---

**Input:** The data matrix $A \in \mathbb{R}^{d \times n}$, target rank $k \in \mathbb{N}$, the number of servers $s \in \mathbb{N}$. The columns of $A$ are assumed to be randomly partitioned among servers $T_1, T_2, \ldots, T_s$.
**Output:** A subset of columns $A_T$ from $A$. $|T| = O(\frac{k}{\sigma})$, where if $A_{OPT}$ is the optimal subset of columns of $A$ of size $k$, and the columns of $A_{OPT}$ are normalized to have unit $\ell_2$ norm, then $\sigma$ is the smallest singular value of $A_{OPT}$.
$S_i \leftarrow$ GREEDY$(A, T_i, \frac{32k}{\sigma})$ for all $i \in [s]$ (The $i^{th}$ server performs this computation.)
Each server sends its $S_i$ to the coordinator.
$T \leftarrow \cup_{i=1}^s S_i$ (This computation is done by the coordinator.)
$S \leftarrow$ GREEDY$(A, T, \frac{12k}{\sigma})$
The coordinator returns $\arg \max_{S' \in \{S, S_1, S_2, \ldots, S_s\}} |\pi_{S'} A|_2^2$

---

In our implementation of Algorithm 8, we make use of Projection-Cost Preserving sketches (PCPs). Another optimization described in Altschuler et al. (2016), which we use in our implementation, is the LAZIER-THAN-LAZY-GREEDY ALGORITHM. The difference between the GREEDY$(A, T, r)$ algorithm and the LAZIER-THAN-LAZY-GREEDY$(A, T, r)$ algorithm is as follows: while at each of the $r$ iterations, GREEDY considers *all* columns of $T$ and chooses the one that leads to the most improvement in the objective, LAZIER-THAN-LAZY-GREEDY samples $\frac{|T| \log(\frac{1}{\delta})}{r}$ columns uniformly at random for some small $\delta$ (which we take in our implementation to be 0.005), and out of those columns, chooses the one which leads to the greatest improvement (where $|T|$ is the number of columns in $T$). This leads to a significant speedup to GREEDY, and it was shown in Altschuler et al. (2016) that this does not significantly worsen the approximation guarantees that Altschuler et al. (2016) shows for GREEDY.

## F.2 SETUP

We compare our distributed protocol, in the case $p = 1$, to Algorithm 8. For both protocols, at the outset we fix the number of columns which are selected. Algorithm 8 is rewritten as Algorithm 9 to reflect this. In this section, we use $k$ to denote the number of columns ultimately selected.

---

**Algorithm 9** Distributed Greedy Column Subset Selection for the Frobenius Norm (Algorithm 2 of Altschuler et al. (2016)). For our empirical comparison, we fix the number of columns selected at the outset.

---

**Input:** The data matrix $A \in \mathbb{R}^{d \times n}$, desired number of columns $k \in \mathbb{N}$, the number of servers $s \in \mathbb{N}$. The columns of $A$ are assumed to be randomly partitioned among servers $T_1, T_2, \ldots, T_s$.
**Output:** A subset of columns $A_T$ from $A$, with $|T| = k$.
$AR \leftarrow$ a PCP for $A$ as discussed in the previous section. All servers and the coordinator have access to $AR$.
$S_i \leftarrow$ GREEDY$(AR, T_i, k)$ for all $i \in [s]$ (The $i^{th}$ server performs this computation.)
Each server sends its $S_i$ to the coordinator.
$T \leftarrow \cup_{i=1}^s S_i$ (This computation is done by the coordinator.)
$S \leftarrow$ GREEDY$(AR, T, k)$
The coordinator returns $\arg \max_{S' \in \{S, S_1, S_2, \ldots, S_s\}} |\pi_{S'}(AR)|_2^2$

---

Note that in Algorithm 9, the coordinator now chooses as many columns as chosen by the servers, as opposed to Algorithm 8, where it chooses a somewhat smaller number of columns — note that this cannot harm the performance of the algorithm.

We compare our protocol to Algorithm 9, using the following datasets:

- gastro_lesions, a $76 \times 698$ matrix dataset available at https://archive.ics.uci.edu/ml/datasets/Gastrointestinal+Lesions+in+Regular+Colonoscopy.

- `secom`, a $591 \times 1567$ matrix dataset. `secom` has missing entries, which we replace with 0s for the purposes of our experiments. `secom` is available at https://archive.ics.uci.edu/ml/datasets/SECOM.

### F.2.1 PARAMETERS

We compare our distributed protocol for $k$-CSS$_1$, using Greedy $k$-CSS$_{1,2}$ as a subroutine, with Algorithm 9, for several choices of $k$ — on `gastro_lesions`, we let $k \in \{10, 20, 30\}$, while on `secom`, we let $k \in \{30, 60, 90, 120\}$. For our protocol, `cauchy_size` is set to $2k$ and `coreset_size` is set to $5k$ for both datasets (where `cauchy_size` and `coreset_size` have the same meanings as in the main body of this paper.) For Algorithm 9, the number of columns in the PCPs is set to $8k$ on `gastro_lesions`, and $7k$ on `secom`.

The hyperparameters `coreset_size` and `cauchy_size`, and the number of columns in the PCPs, are set this way so that the amount of communication that each algorithm is allowed is roughly equal (Algorithm 9 is allowed slightly more communication). To see this, let $d$ be the number of rows in the data matrix $A$. If $c$ is the number of columns in the PCPs, then the total communication used to transmit the PCPs between servers is $2scd$, since the servers must first send their respective $A_i R_i$ to the coordinator, and the coordinator then sends $AR$ to all of the servers. By comparison, if $r_1$ is the number of rows in the initial Cauchy matrix in our protocol, and $r_2$ is the number of columns in each coreset sent by the servers to the coordinator, then the total communication required to transmit the Cauchy matrix and the coresets is $(r_1 + r_2)sd$. We choose $r_1, r_2$ and $c$ so that $2scd$ is slightly higher than $(r_1 + r_2)sd$.

Note that if $k$ is the number of columns ultimately selected, then our protocol uses $2dk$ additional bits of communication between the servers and the coordinator to recover the final $k$-subset of columns, while Algorithm 9 will use $sdk$ communication to send the subsets of columns $S_i$ (of size $k$) from the servers to the coordinators. This is not included in our hyperparameter calculations.

### F.2.2 HOW TRIALS ARE CONDUCTED

With these choices of hyperparameters, we conduct 15 trials as follows. In each trial, if $A \in \mathbb{R}^{d \times n}$ is our data matrix, then we shuffle the columns and then divide them equally between 2 servers (since for the theoretical guarantees of Algorithms 8 9 to apply, the columns should be partitioned randomly). Using this partition across 2 servers, we run our protocol and Algorithm 9. For each protocol, once the subset of columns is computed, we perform multiple-response $\ell_1$ regression to evaluate the error.

In the next section, we report the minimum error across the 15 trials for each dataset and for each value of $k$. We also report the mean error, along with the standard deviation. Finally, we compute the work and the span of our protocol and Algorithm 9 using Python's `time.process_time()` utility. This does not include the time taken to perform multiple-response $\ell_1$ regression.

For `secom`, with $k = 30$ and $k = 60$, the trials were performed on a Late-2016 Macbook Pro with a 2.7 GHz Quad-Core Intel Core i7 processor and 16GM of memory. The rest were performed on a 2019 MacBook Pro with a 2.4 GHz Intel Core i5 processor and 8 GB 2133 MHz LPDDR3 memory.

### F.3 RESULTS

Our results are shown and discussed in Figures 3 (minimum and mean/standard deviation for $\ell_1$ error) and 4 (average work and span across 15 trials).

## G FULL EXPERIMENTAL RESULTS

For our protocol, we considered various additional hyperparameter settings on real-world datasets (`bcsstk13`, `isolet` and 5 images in the `Caltech-101` dataset) — these settings are shown in Tables 3, 4, and 5. As before, `cauchy size` is the number of rows in our initial Cauchy matrix, sent to the servers at the beginning of the protocol, and `coreset size` is the size of the strong coreset sent by each server to the coordinator. For regular $k$-CSS$_{1,2}$, `sketch size` is the number of rows in the sparse embedding matrix, and `sparsity` is the number of nonzero entries in each column of the sparse embedding matrix.

For each of these settings, we display the minimum error, as well as the mean error and the standard deviation for each setting and each rank, as shown in Figure 5. We also display the average work/span of each setting, for each rank, as shown in Figure 6. We observe that not only does greedy $k$-CSS$_{1,2}$ perform better than all settings of regular $k$-CSS$_{1,2}$ in minimum and mean errors across multiple trials, it also has a smaller variance in performance.

### G.1 DETAILS OF WORK/SPAN COMPUTATION

In Figure 6, work and span were recorded as the time taken (using Python's `time.process_time()` utility) to compute the column subset in our distributed protocol. In particular, it does not include the time spent performing $\ell_1$ regression to obtain the right factor and consequently the entry-wise $\ell_1$ errors.

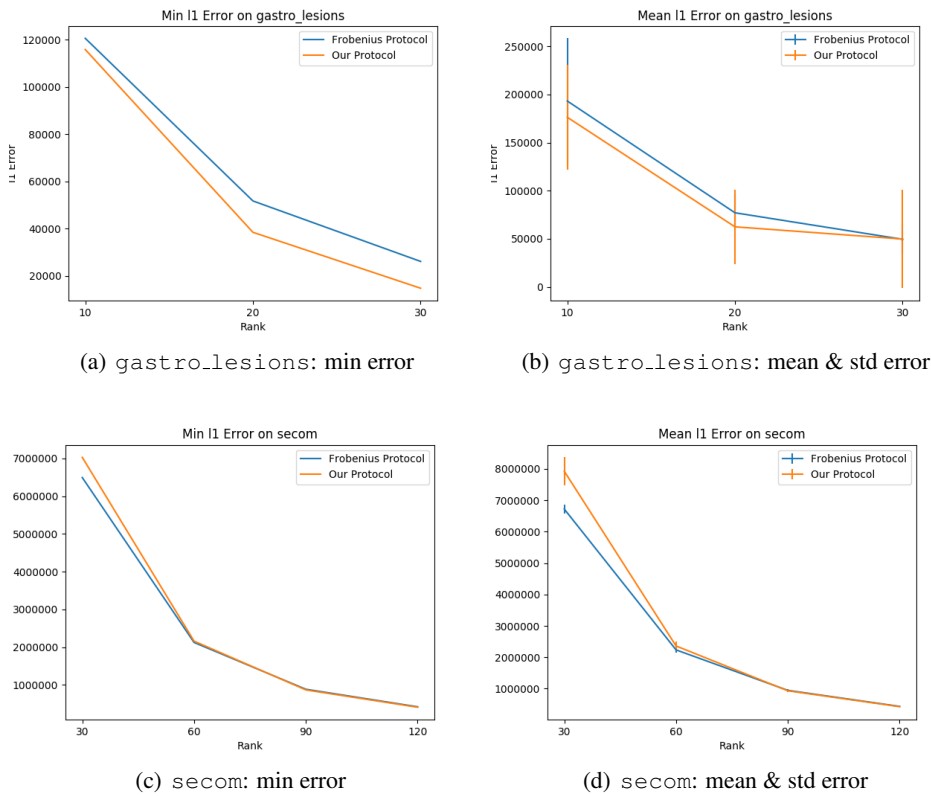

(a) `gastro_lesions`: min error   (b) `gastro_lesions`: mean & std error

(c) `secom`: min error   (d) `secom`: mean & std error

Figure 3: Plots (a) and (c) show minimum $\ell_1$ error across 15 trials on `gastro_lesions` and `secom` respectively, while plots (b) and (d) show the mean $\ell_1$ errors and the standard deviation. For ranks 90 and 120, our protocol affords a $3\%$ improvement in minimum error `secom`, while for rank 30 on `gastro_lesions`, our protocol gives over $40\%$ improvement. Note that the mean error for both protocols is noticeably higher on `gastro_lesions` — in the case of our protocol, the error in the case $k = 30$ is distorted by two trials with $\ell_1$ errors 205827 and 134945 respectively.

| Setting Number | 1 | 2 | 3 | 4 | 5 | 6 | 7 | 8 | 9 | 10 |
|---|---|---|---|---|---|---|---|---|---|---|
| sketch size | $3k$ | $5k$ | $3k$ | $5k$ | $k/2$ | $k/3$ | $k/5$ | $k/2$ | $k/3$ | $k/5$ |
| sparsity | $\min(20,3k)$ | $\min(20,5k)$ | $\min(40,3k)$ | $\min(40,5k)$ | $\min(2,k/2)$ | $\min(2,k/3)$ | $\min(2,k/5)$ | $\min(5,k/2)$ | $\min(5,k/3)$ | $\min(5,k/5)$ |

Table 3: All hyperparameters used on `bcsstk13`, when regular $k$-CSS$_{1,2}$ is used. In this table, $k$ denotes the number of columns ultimately selected. In all settings (including when greedy $k$-CSS$_{1,2}$, not included here, is used), `cauchy size` is either $5k$ or $8k$, and `coreset size` is $5k$. The setting numbers are shown on top — in the following error plots, Setting 0 is used to refer to our protocol when greedy $k$-CSS$_{1,2}$ is used. Note that `sparsity` can be at most `sketch size`, since `sketch size` is the number of rows of the sparse embedding matrix, while `sparsity` is the number of nonzero entries in any column. (There is a slight typo in Table 2 in Section 8 of our main paper, where `coreset size` is given as $10k$.)

| Setting Number | 1 | 2 | 3 | 4 | 5 | 6 | 7 | 8 | 9 | 10 |
|---|---|---|---|---|---|---|---|---|---|---|
| sketch size | $3k$ | $5k$ | $3k$ | $5k$ | $k/2$ | $k/3$ | $k/5$ | $k/2$ | $k/3$ | $k/5$ |
| sparsity | $\min(20,3k)$ | $\min(20,5k)$ | $\min(40,3k)$ | $\min(40,5k)$ | $\min(2,k/2)$ | $\min(2,k/3)$ | $\min(2,k/5)$ | $\min(5,k/2)$ | $\min(5,k/3)$ | $\min(5,k/5)$ |

Table 4: All hyperparameters used on `isolet`, when regular $k$-CSS$_{1,2}$ is used. In all settings (including when greedy $k$-CSS$_{1,2}$, not included here, is used), `cauchy size` is $4k$, and `coreset size` is $4k$. The setting numbers are shown on top — in the following error plots, Setting 0 is used to refer to our protocol when greedy $k$-CSS$_{1,2}$ is used.

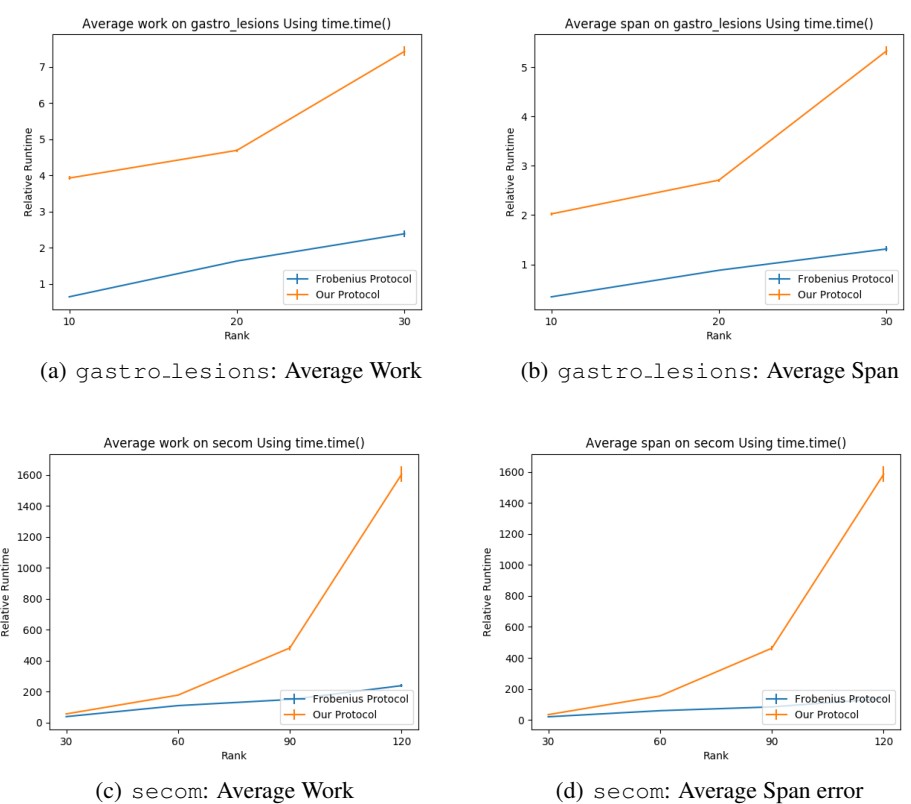

(a) gastro_lesions: Average Work

(b) gastro_lesions: Average Span

(c) secom: Average Work

(d) secom: Average Span error

Figure 4: Plots (a) and (b) show the average work and span respectively for gastro_lesions, while plots (c) and (d) show the average work and span for secom. As expected, our protocol using GREEDY $k$-CSS$_{1,2}$ takes more time — to our knowledge, there is not yet an optimization similar to LAZIER-THAN-LAZY GREEDY for the $\ell_{1,2}$-norm. Nevertheless, running time is less important than communication in the distributed setting.

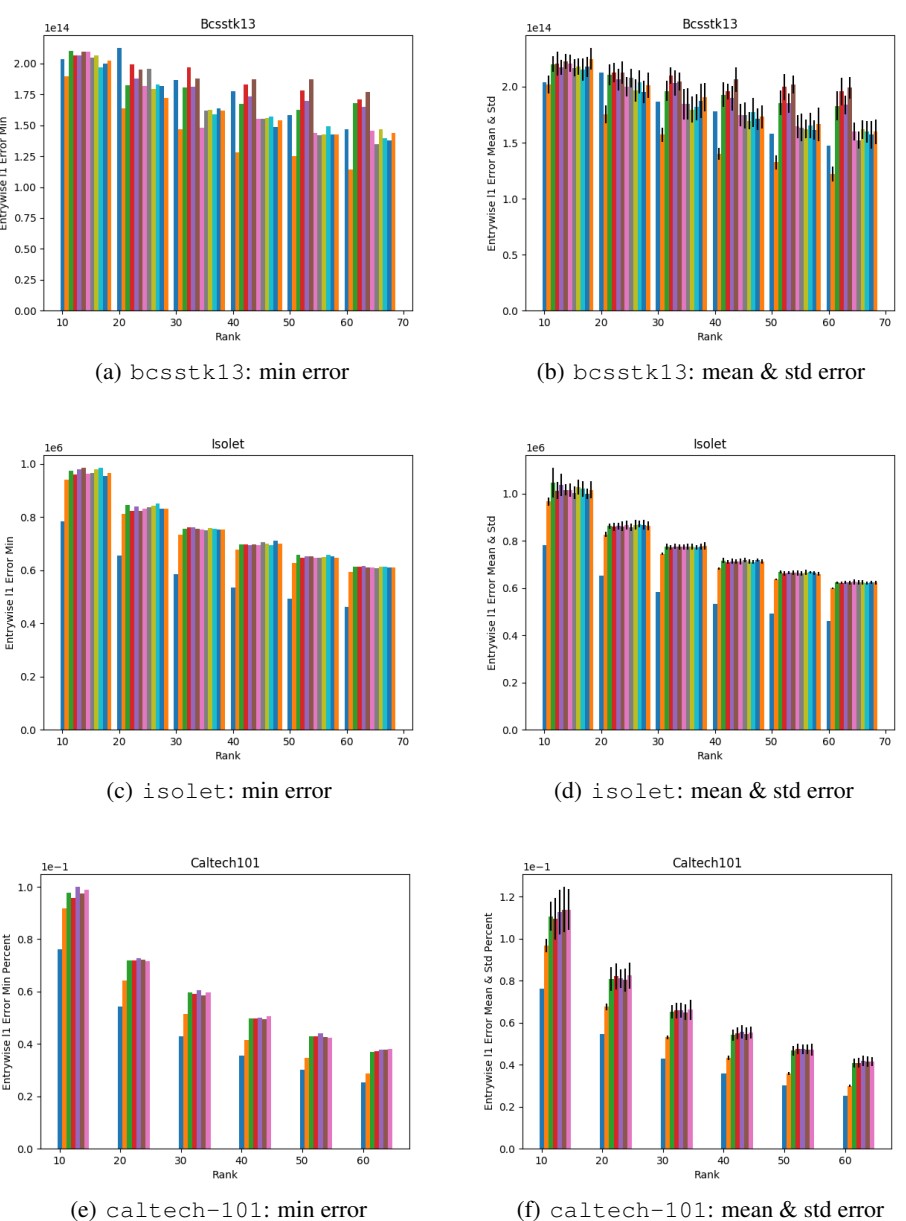

Figure 5: Results on bcsstk13, isolet, and caltech-101 from top to bottom. The left plots show minimum error across all 15 trails; the right plots show the corresponding mean and standard deviation in error. In all plots, the first bar denotes SVD, the second bar denotes greedy $k$-CSS$_{1,2}$, and the rest of the bars denote all settings of regular $k$-CSS$_{1,2}$, at all ranks on the axis 10 through 60.

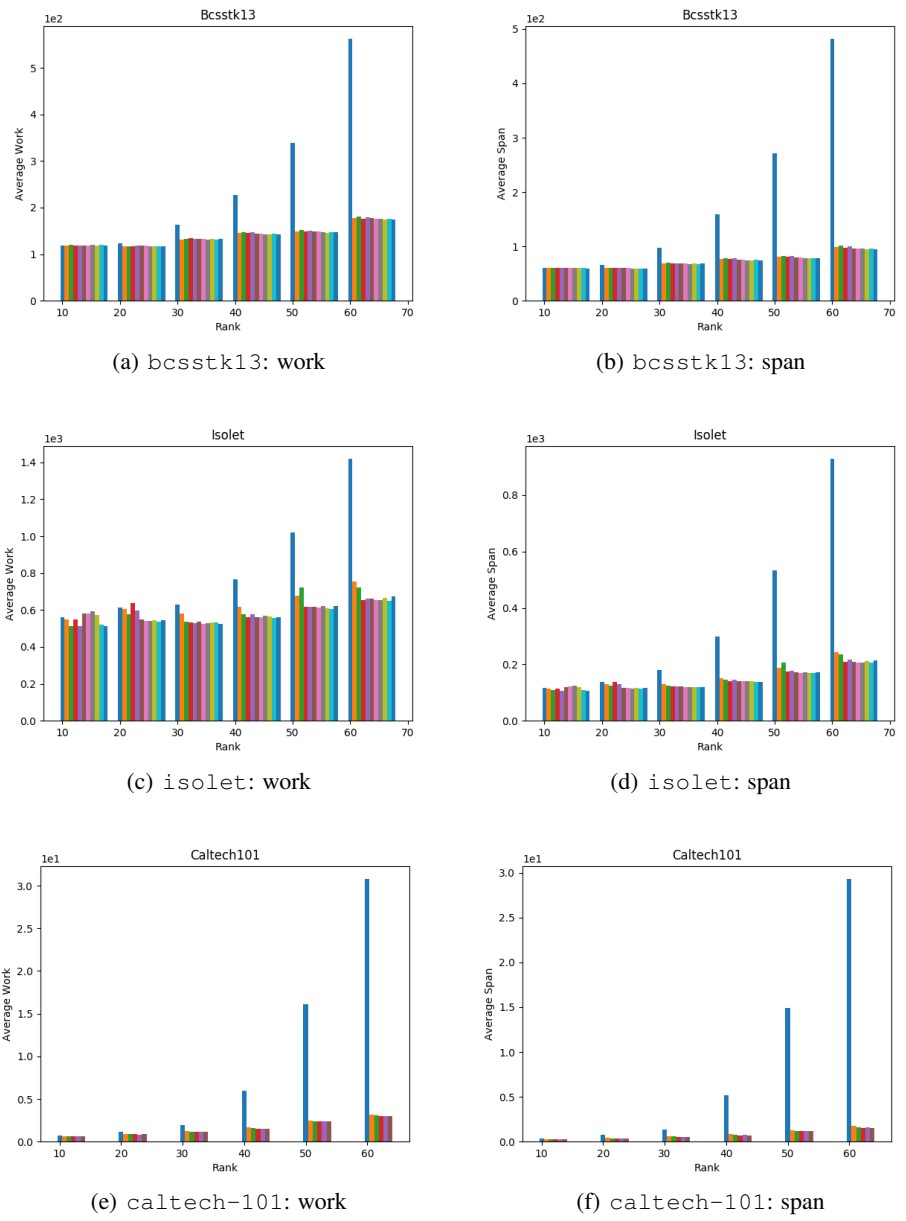

Figure 6: Results on bcsstk13, isolet, and caltech-101 from top to bottom. The left plots show average work in seconds across all 15 trails; the right plots show the corresponding average span. In all plots, the first bar denotes greedy $k$-CSS$_{1,2}$, and the rest of the bars denote all settings of regular $k$-CSS$_{1,2}$, at all ranks on the axis 10 through 60.

| Setting Number | 1 | 2 | 3 | 4 | 5 |
|---|---|---|---|---|---|
| sketch size | $k$ | $k/3$ | $k/5$ | $k/3$ | $k/5$ |
| sparsity | $\min(20, k)$ | $\min(2, k/3)$ | $\min(2, k/5)$ | $\min(5, k/3)$ | $\min(5, k/5)$ |

Table 5: All hyperparameters used on `isolet`, when regular $k$-CSS$_{1,2}$ is used. In all settings (including when greedy $k$-CSS$_{1,2}$, not included here, is used), `cauchy size` is $4k$, and `coreset size` is $4k$. The setting numbers are shown on top — in the following error plots, Setting $0$ is used to refer to our protocol when greedy $k$-CSS$_{1,2}$ is used.

Figure 5 shows that we encounter a tradeoff between accuracy and running time when choosing between these two subroutines. Both lead to the same overall communication cost, and since accuracy is of more interest than running time in the distributed setting, it is (empirically) preferable to use Greedy $k$-CSS$_{1,2}$ within the protocol.

### G.2 ADDITIONAL DETAILS

All experiments on the `Caltech-101` dataset were run on a Late-2016 Macbook Pro with a 2.7 GHz Quad-Core Intel Core i7 processor and 16GM of memory. All experiments on `bcsstk13` and `isolet` were run on an AWS z1d.xlarge instance with Deep Learning AMI (Amazon Linux 2) Version 29.0.

