# OpenReview forum: "An Efficient Protocol for Distributed Column Subset Selection in the Entrywise $\ell_p$ Norm"
_ICLR.cc/2021/Conference — Reject_

### Official Review · AnonReviewer4 · 2020-10-24
**Review of An Efficient Protocol for Distributed Column Subset Selection in the Entrywise lp Norm**

**Rating:** 7
**Confidence:** 3

**Review:**

SUMMARY:

The paper considers the column subset selection (CSS) problem, which has received considerable attention in numerical linear algebra. It considers a distributed variant of CSS in the $\ell_p$ norm, where $p \in [1,2)$. Despite the attention this problem has received previously, it seems like this is a new setting that has not been considered before. The paper primarily provides theoretical results, but also does some experiments.

I think this is an interesting paper and problem. However, there are a few things I'd like to see addressed before I can give a higher score. There are a few details in the proof of Theorem 6 that are unclear. Moreover, the readability of the paper could be improved. I provide further details below.


ADVANTAGES:

- CSS is an important problem in numerical linear algebra.
- The setting under consideration seems to be new.
- The paper provides both theoretical and experimental results.


CONCERNS/QUESTIONS:

- Section 1 would be easier to follow if the definitions in Section 2.1 were given when the concepts are first used in Section 1 instead.

- The p-stable distribution is discussed throughout the paper, but doesn't seem to be defined anywhere, not even in the supplement. It would be helpful if the p-stable distribution was defined when it is first mentioned. Also, in the proof of Lemma 3.1, it would be helpful if you could point to a reference that contain the relevant facts about p-stable distributions.

- In Section 1.2, the first paragraph, you make a distinction between an approximation to OPT and an approximation to the best possible subset of columns. Can you clarify the distinction between these two types of approximation?

- The numbering of the lemmas is confusing. The numberings used in the main paper are reused in the supplement, but for different results. For example, Lemma 1 in the main paper is called Lemma 2 in the supplement, and Lemma 2 in the main paper is called Lemma 3 in the supplement. This makes the proof of Theorem 6 confusing.

- On the 4th line in Section 1.1., you say that the coordinator computes $\sum_i A_i S = A S$. But $A$ is of size $d \times n$, and each $A_i$ is of size $d \times n_i$, so this equality isn't right. Do you mean to say that $\sum_i \tilde{A}_i S = A S$, where each $\tilde{A}_i$ is of size $d \times n$ and with all entries zero except those columns corresponding to the columns in $A_i$ (or something to that effect)?

- In the proof of Theorem 6, when you apply Lemma 5 the first time: Don't you need both $V'$ and $M$ to be defined in terms of the $p,2$-norm rather than the $p$ norm for Lemma 5 to apply? In other words, should they be defined as

$V' := \arg \min_V |S A_I V - S A|_{p,2}$

and

$M := \arg \min_M | S A_I M - S A T |_{p,2}$?

- In the proof of Theorem 6, when you apply Alg. 4: It is not clear how Alg. 4 from the supplement is applied here. Alg. 4 outputs factors U, V such that UV approximates A. What is $(SAT)^*$? Is $(SAT)^* = UV$ here?

- In the proof of Theorem 6, when you apply Lemma 5 the second time: It looks like this uses the first equality in Lemma 5 in the supplement, which requires $P_2^*$ to be a projection matrix. How can we know that it is? Moreover, even if the sampling matrices $T_i$ are chosen to satisfy Lemma 5, how do we know that the matrix $T$ obtained from the $T_i$'s in Alg. 1 also satisfies Lemma 5?

- What is the run time of Algorithm 2? Does finding each $j^*$ require evaluating the objective for every choice of $j \in \overline{T}$, or is there a better way to do this?

- In Section 6, in the Setup paragraph, you say that you report the minimum error over 15 trials. Is there a reason for reporting the minimum rather than the mean or median? Is the mean/median also competitive?

- The paper ends abruptly with no conclusion.


MINOR CONCERNS/QUESTIONS:

- In the sentence right after Lemma 2 in the main paper, should $|S A_T V - S A_T|_p$ be $|S A_T V - S A|_p$?

- In Algorithm 2, should the argmin be over $j \in \overline{T}$ rather than $j \in A_{\bar{T}}$?

- In Tables 3, 4 and 5 of the supplement, should k-CSS_{!,2} be k-CSS_{1,2} (i.e., a '1' instead of an exclamation point)?


###################################

Update:

The authors have addressed the concerns I had in my initial review. I have raised the score from 6 to 7.

---

> ### Author Response · Authors · 2020-11-25
> **Thank you for your comments.**
>
> We thank the reviewer for the valuable feedback. We address the concerns as follows.
>
> 1. In Section 1.2, the first paragraph, you make a distinction between an approximation to OPT and an approximation to the best possible subset of columns. Can you clarify the distinction between these two types of approximation?
>
> Approximation to the best possible subset of columns means an upper bound on $\min_{A_L, V} |A_LV - A|_p$, where $A_L$ is a subset of $O(k)$ columns of the original matrix $A$. Approximation to OPT means an upper bound on $\min_{\textrm{rank-k}\ A_k} |A_k - A|_p$, which is the lowest possible cost of approximation for any rank-$k$ matrix $A_k$. Note that although both $A_LV$ and $A_k$ are rank-$k$ approximations to $A$, there is no restriction on the matrix $A_k$ in approximation to OPT.
>
> 2. The numbering of the lemmas is confusing. The numberings used in the main paper are reused in the supplementary, but for different results. For example, Lemma 1 in the main paper is called Lemma 2 in the supplementary, and Lemma 2 in the main paper is called Lemma 3 in the supplementary. This makes the proof of Theorem 6 confusing.
>
> We fixed the numbering of the lemmas in the main paper and in the Appendix. They now agree with each other.
>
> 3. On the 4th line in Section 1.1., you say that the coordinator computes $\sum_i A_i S = AS$. But $A$ is of size $d \times n$, and each $A_i$ is of size $d \times n_i$, so this equality isn't right. Do you mean to say that $\sum_i \widetilde{A_i}S = AS$, where each $\widetilde{A_i}$ is of size $d \times n$ and with all entries zero except those columns corresponding to the columns in $A_i$ (or something to that effect)?}
>
> Sorry for the confusion on $\sum_{i} A_iS = AS$. Yes, we mean a column-wise concatenation of $A_iS$'s. This has been updated on the 4th line in Section 1.1.
>
> 4. In the proof of Theorem 6, when you apply Lemma 5 the first time: Don't you need both $V'$ and $M$ to be defined in terms of the $p, 2$-norm rather than the $p$ norm for Lemma 5 to apply? In other words, should they be defined as $V' := argmin_V |SA_I V - SA|_{p, 2}$ and $M := argmin_M |SA_IM - SAT|_{p, 2}$
>
> Thanks - we updated the proof of Theorem 6. We removed the confusing steps in the analysis. Specifically, $V' = argmin_V |SA_IV - SA|_{p, 2}$. $(SAT)^*$ is the best rank-$k$ approximation to $SAT$.
>
> 5. In the proof of Theorem 6, when you apply Alg. 4: It is not clear how Alg. 4 from the supplement is applied here. Alg. 4 outputs factors U, V such that UV approximates A. What is $(SAT)^*$? Is $(SAT)^* = UV$ here?}
>
> See the answer to the previous question.
>
> 6. In the proof of Theorem 6, when you apply Lemma 5 the second time: It looks like this uses the first equality in Lemma 5 in the supplementary, which requires $P_2^*$ to be a projection matrix. How can we know that it is? Moreover, even if the sampling matrices $T_i$ are chosen to satisfy Lemma 5, how do we know that the matrix $T$ obtained from the $T_i$'s in Alg. 1 also satisfies Lemma 5?}
>
> For the strong coreset introduced in Lemma 5, we removed the projection statement, which is not necessary in the analysis for Theorem 6.
>
> In the new analysis, we make the step that applies Lemma 5 (strong coreset) clear. Specifically, we apply Lemma 5 to each individual coreset $SA_iT_i$ and show that $SAT = [SA_1T_1, \dots, SA_sT_s]$ preserves the cost of subspace projection for $SA = [SA_1, \dots, SA_s]$ in the $\ell_{p, 2}$ norm.
>
> 7. What is the running time of Algorithm 2? Does finding each $j^*$ require evaluating the objective for every choice of $j \in \overline{T}$, or is there a better way to do this?}
>
> Thank you for the suggestion, it is not necessary to evaluate the objective for every single choice of $j \in \overline{T}$. Instead, inspired by the Lazier-than-lazy heuristic used by Altschuler, et al. (2016), we can sample $\frac{n \log(1/\delta)}{k}$ of the remaining column indices $j$ of $A$ (where $\delta$ is some small parameter), and we only have to evaluate the objective for each of those column indices $j$, rather than for all the remaining columns of $A$. See Section 5 of our updated version.
>
> 8. In Section 6, in the Setup paragraph, you say that you report the minimum error over 15 trials. Is there a reason for reporting the minimum rather than the mean or median? Is the mean/median also competitive?
>
> We reported the mean error for all of the settings as well, in Figure 5 in Appendix G. The mean is competitive as well. We reported the mean for our updated experiments.
>
> We have addressed all presentation-related comments as well, thanks.

---

### Official Review · AnonReviewer2 · 2020-10-26
**above acceptance threshold**

**Rating:** 6
**Confidence:** 4

**Review:**

This paper focused $\ell_p$-norm error analysis of distributed column subset selection problems. The proposed distributed algorithm guarantee an $\tilde O(k^{1/p-1/2} )$-approximation to the best subset of columns with $\tilde O(sdk)$ communication cost per round and polynomial time.  The lower bound analysis show that such communication complexity is near optimal. The design and analysis of the algorithm is based on f oblivious sketching and strong coresets, which leads to the overall approximation factor is only $\tilde O(k^{1/p-1/2} )$. The authors also extended the main idea to address $\ell_{1,2}$ norm error.

I have not checked all details of the proof, but the results and the main ideas look reasonable. I think the theoretical contribution of this paper is valuable, however, the contents is somewhat limited to the audience of ICLR. There are some questions/comments:

1. I strongly encourage the authors apply the proposed algorithm to address the real-world problems such as computer vision, image processing which is mentioned in introduction.

2. Since the column select problem focus on the matrix $A\in\mathbb{R}^{d\times n}$ with $n\gg d$, the empirical studied should also contain such case. It is observed that proposed method performs better than SVD on both synthetic dataset and some real world dataset. I think one of the reason may be the $l_p$ norm for $1\leq p<2$ is not a unitary invariant norm and SVD do not give an optimal low-rank approximation. Hence, SVD may be not a good deterministic algorithm as baseline.

3. The presentation could be improved: a) The operations on Server and coordinates should be described in multiple lines.  b) The curves in figure 2 should be displayed with different types of lines.

---

> ### Author Response · Authors · 2020-11-25
> **Thank you for your comments.**
>
> We thank the reviewer for the valuable feedback. We address the concerns as follows.
>
> 1. We updated the experimental section with a real-world application, term-document clustering, where previous $k$-CSS$_2$ algorithm is applied (``CUR matrix decompositions for improved data analysis'' by Michael W. Mahoney and Petros Drineas).
>
> 2. The experiment section now includes empirical study on a dataset of size 139 by 18446, which fits our setting well. There is no widely applied baseline for $\ell_1$ low rank approximation. The $\ell_1$ low rank approximation algorithms are usually compared against SVD, for example, in paper "Towards a zero-one law for column subset selection", "Low Rank Approximation with Entrywise $\ell_1$-Norm Error". Furthermore, as discussed in the introduction, extending existing single machine  $k$-CSS$_1$ into a distributed setting would require $O(\log n)$ rounds. What is worse, in each round, it needs to compute an $\ell_1$ regression on all columns of the entire data matrix to determine which columns are well-approximated by the selected columns. This can take a prohibitively long amount of time, and thus is not a good baseline for our empirical study.
>
> 3. We modified our presentation accordingly, thanks.

---

### Official Review · AnonReviewer3 · 2020-10-28
**Single Round Distributed CSSP**

**Rating:** 5
**Confidence:** 4

**Review:**

The authors present a distributed protocol for the column subset selection problem under the $\ell_p$ norms for $1\leq p <2$. The model of computation is a standard coordinator model of communication where the input matrix A is column partitioned to $s$ servers. The main contribution consists of a protocol that requires $O(sdk)$ communication, single round and polynomial running time and returns a $k^{1/p - 1/2}$ multiplicative approximation to the best possible column subset.

#### Reasons for score:
Overall, I vote for a borderline / slightly below acceptable but I am open to discussion about my decision. My main concerns are (a) the polynomial time complexity of the method makes the proposed protocol hard to be useful in real applications, and (b) the experimental evaluation is done on very small matrices (i.e., 2000 x 2000 shape) as well as small number of servers. On the positive side, the paper is a solid theoretical work on the distributed CSSP problem.

#### Strong points:
* Well written paper with a clear contribution statement
* Related work is up-to-date (to the best of my knowledge)
* Concise algorithm description and corresponding theoretical guarantees.


#### Concerns:
* The polynomial running time guarantee makes the practicality of the proposed algorithm marginal. Do you (implicitly) assume that you have limited space per server as well? Please make such assumption explicit.
* Experimental section need sto be extended with larger datasets and more servers. How does the distributed protocol compares with a centralized one? The size of the datasets here suggest the following question.
* Why is the distributed setting is studied here? If the resulting algorithm is polynomial, why not solving the problem in a single server?

Minor comments:
* Explicitly state what is hidden under the notation $\tilde{O}$.
*Lemma 3: the probability constant $0.999$ seems arbitrary here, please replace it with $0<\delta<1$ failure probability parameter.
* Lemma $4$: What is a "sampling and reweighting matrix T"? Is this defined in the main text?
* After Theorem 5: give a reference to "Lewis weights".
* Before Section 5 (Runtime): Why generating a p-state random variable takes constant time?
* Is all communication bounds in bits?
* Figure 2 is not easy to read, please make it larger and increase the font on both axis.

---

> ### Author Response · Authors · 2020-11-25
> **Thank you very much for your comments.**
>
> We thank the reviewer for the valuable feedback. We address the major concerns as follows.
>
> 1.  We update the running time analysis in the paper and make it explicit what all of the ``polynomial'' running times are. We do not apply implicit assumptions that each server has limited space. However, we do consider limited communication cost between the servers and the coordinator.
>
> 2. We updated the experimental section with one real-world application on larger a dataset of size 139 by 18446 and with 20 servers. We compare our distributed protocol against a centralized SVD baseline.
>
> 3. We added a stronger motivation to study $k$-CSS in a distributed setting at the beginning of paragraph 3 in the Introduction. It is important to consider distributed $k$-CSS because:
>
>     1) The dataset might be too large to store on a single server.
>
>     2) The dataset might be so large that it takes prohibitively long for a single server to select the columns. For example, using the best single machine $k$-CSS$_1$ algorithm from (``Low Rank Approximation with Entrywise $\ell_1$-Norm Error" by Song, Woodruff, and Zhong), to select even 10 columns from a dataset $A \in \mathbb{R}^{800 \times 20000}$ would take more than one day, because the algorithm involves computing the Lewis weights for 20000 columns. However, in a distributed setting, we can distribute the 20000 columns to 20 servers, so that each server only holds 1000 columns. Our protocol uses the Lewis weights to select a subset of (reweighted) columns (a strong coreset) on each server, which preserves the approximation cost up to $(1\pm \epsilon)$ w.h.p. Computing the Lewis weights for 1000 columns is now significantly faster than before, and all the servers can compute the Lewis weights and then select the columns in parallel.
>
>     3) The data is collected at multiple places and it is not feasible to transfer all the data to a central server, which incurs a high communication cost.
> \end{enumerate}
>
> 4. We addressed all remaining comments. In Section 2.2, we added a more detailed description of $p$-stable distributions and explicitly state why generating a $p$-stable random variable takes constant time. For our communication bounds, as stated in Section 2.1 where the column partition model is introduced, ` the communication cost is the total number of words transferred between the servers and the coordinator over the course of the protocol. Each word is $O(\log(snd))$ bits''.

---

### Author Response · Authors · 2021-02-24
**Improvements We Made to Our Paper**

Thank you very much for your valuable feedback. We have made the following major updates to our main paper.

- We give a single-pass streaming algorithm based on our techniques as our main result and applications of our techniques now. We emphasize the importance of single-pass streaming algorithms, as the problems become trivial if enough ($\log n$ passes) or space is allowed for the streaming algorithm. The problem is extensively studied for other norms, such as Frobenius, but not for entrywise $\ell_p$-norms for $p \neq 2$.
- We more clearly detail our technical novelties for both the streaming algorithm and the distributed protocol in the introduction.
- We updated the experiments, applying both of our streaming algorithms and our distributed protocol to synthetic data and two real-world applications on text data and genetic analysis (one sparse data set and one dense data set). We made the implementation of our algorithms more scalable, using Python Ray, a high-level distributed and parallel computing framework. All of our code will be open-sourced.
- In addition to comparing the $\ell_1$ approximation error of each algorithm, we also include comparisons of the wall-clock time that each algorithm uses to select our subset of columns.
- In the experiments, we compare one of our $k$-CSS$_{1, 2}$ algorithms and one Frobenius norm $k$-CSS$_2$ algorithm with a random baseline to select noisy images for classification. This demonstrates the robustness of the $\ell_1$ norm over the Frobenius norm for $k$-CSS.

---

### Decision · Program_Chairs · 2021-01-07
**Final Decision**

**Decision:**

Reject

**Comment:**

Clarity: Well written paper with a clear contribution statement; related work is up-to-date; concise algorithm description and corresponding theoretical guarantees. However, the presentation could be still improved.

Significance: The polynomial running time guarantee makes the practicality of the proposed algorithm marginal. Experimental results do not back up strongly the significance of the algorithm.

Main pros:
- Solid theoretical work on the distributed CSSP problem.

Main cons:
- The reviewers point out the significance of the experimental results, beyond the theoretical contribution. For the ICLR audience, real, large-scale experiments, that really dictate that the proposed solutions is (if not the only) one of the few solutions to follow, are necessary. The reviewers highlight that the theoretical results need to be applied to really large-scale scenarios (e.g., the problems considered in the paper can definitely be handled by a single computer, and no distributed implementation is required).
- Τhe polynomial time complexity of the method makes the proposed protocol hard to be useful in real applications.
- How does the distributed protocol compares with a centralized one? This is not fully addressed in the rebuttal.